# On Local Policies for Graph-Structured Markov Decision Processes

**Fathima Zarin Faizal** [1]   **Asuman Ozdaglar** [1]   **Martin J. Wainwright** [1]

## Abstract

We study a cooperative form of multi-agent reinforcement learning with state space dynamics and agent interaction controlled by an underlying graph. Each agent has a local state and action, the evolution of the local state depends only on the states and actions in the 1-hop neighborhood defined by the graph. Structured dynamics of this type arise in various applications, including network resource allocation, co-operative games, epidemic control, and wireless scheduling. The global state-action space scales exponentially in the number of agents, so that computing global optimal policies is intractable in the worst-case. We study conditions under which it is possible to approximate the optimal policies by a local policy for each agent that depends only on states associated with nodes within its $m$-hop neighborhood. By controlling the propagation of influences via a Dobrushin-type stability matrix, we establish that globally optimal policies can approximated by local policies with sub-optimality gap decaying exponentially in $m$.

## 1. Introduction

Markov decision processes (MDPs) provide a formalization of sequential decision-making under uncertainty, and have proven useful in a wide variety of applications. Optimal policies are characterized by the Bellman principle, and in theory, can be computed either via exact dynamic programming when the model is known (e.g. (Puterman, 2014; Bertsekas, 2025)), or approximated using samples via reinforcement learning (RL) algorithms (Bertsekas, 2025; Sutton & Barto, 2018). In either setting, the curse-of-dimensionality manifests itself via growth of the state and/or action spaces. Accordingly, there is a large body of work in approximate

dynamic programming and/or RL that focuses on particular structured classes of MDPs, and associated schemes for computing near-optimal policies.

In many application domains, there exists additional structure in the form of an underlying graph. This is true, for instance, in controlling epidemics and wildfires (Peyrard et al., 2007; Haksar & Schwager, 2018); wireless scheduling (Feriani & Hossain, 2021); and controlling robots and autonomous vehicles (Kober et al., 2013; Shalev-Shwartz et al., 2016). In this paper, we study MDPs with state and action spaces that decompose according to an underlying graph, and for which the state dynamics respect the graph structure. Associated with each node is an agent, along with a local state, a local action, and a local reward function that depends only on the local state-action pair. Moreover, any agent's actions have a direct effect only on its own states, and those of its immediate neighbors, and the state dynamics depend only on the local neighborhood of state-action pairs. We study the cooperative setting, in which agents share the common goal of maximizing the discounted long-term reward summed across all of the agents.

One concrete instantiation is the problem of epidemic control (Peyrard et al., 2007; Haksar & Schwager, 2018). In this setting, each agent corresponds to a person, and the local state could be a binary variable indicating whether or not they are infected. An agent can take a range of possible actions, which (depending on the disease) could include taking medicine, vaccination, or self-isolation. Edges in the graph correspond to dependencies between individuals' infection statuses; the probability of transition of a particular individual's infection status directly depends on their neighborhood's infection status and the control action taken. See Section 5 for further discussion of this example.

Now consider the problem of finding an optimal policy. A government agency might seek to maximize the long-term health of all the individuals, but it is infeasible to have global co-ordination among all individuals, along with disclosure of their private health information. Consequently, it is natural to seek policies that are $m$-hop local, meaning that every individual can make decisions based on information that depends only on individuals at distance at most $m$ in the underlying graph.

For a large-scale networked system, it becomes intractable

---

[1]Laboratory for Information and Decision Systems, EECS, Massachusetts Institute of Technology. Correspondence to: Fathima Zarin Faizal <fathima@mit.edu>.

*Proceedings of the 43${}^{rd}$ International Conference on Machine Learning*, Seoul, South Korea. PMLR 306, 2026. Copyright 2026 by the author(s).

for any single agent, with limited memory and/or computational power, to implement a globally optimal policy. Thus, one is led to the following natural question:

> Under what conditions can the globally optimal policy of an agent be well-approximated by a local policy, depending only on a local neighborhood of states and actions?

The main contribution of this paper is to study this question—and to provide some precise answers—for this broad class of networked multi-agent systems.

**Our contributions.** The main contribution of this paper is to give sufficient conditions for the existence of $m$-hop local policies that are good approximations to the globally optimal policy. In particular, our results show that the existence of near-optimal local policies depends on the *propagation of influences* in the underlying network MDP. In this way, we establish connections between approximation schemes for networked MDPs, correlation decay in graphical models (Gamarnik et al., 2014), and mixing time analysis of Markov chains (Levin & Peres, 2017).

Obtaining our theoretical guarantees involves tackling two broad challenges, both of which play an important role in the multi-agent RL literature. The first is that of *partial observability*: if agents do not use the entire state-space, how much do they lose? The second is that of *coordination* between the agents: how can they communicate with their neighborhood to collaboratively compute and implement an approximately optimal policy?

1. To focus on the effect of partial observability, we first study a simpler setting, involving a single active agent, where no coordination is required. Here our main result (Theorem 1) gives conditions under which there exists a $m$-local policy with sub-optimality gap that decays exponentially with $m$. Moreover, we describe how to compute this local policy in a space- and time-efficient manner.

2. Using insights from controlling partial observability errors in the single-agent case, we then turn to the general multi-agent problem. In Theorem 2, we show the existence of an approximately-optimal policy that only requires each agent to communicate and coordinate with their $m$-hop neighborhood, with a suboptimality gap that decays exponentially in $m$. The structure of the policy shows how agents can coordinate their actions to achieve approximately-optimal performance.

In both of our results, we provide explicit bounds on the sub-optimality gap relative to the $Q$-function of the globally optimal policy. This type of guarantee differs from past work that either restricts to the class of $0$-hop local policies (Qu et al., 2020), or compares to an entropy-regularized policy (Zhang et al., 2023). To the best of our knowledge, our results are the first to establish the existence of near-optimal local policies, as measured relative to the global optimum. In certain versions of the multi-agent set-up, our policies require agents to coordinate their actions with their neighbors. Such local co-ordination policies have not been studied before in the literature, and suggest multiple open directions for future work.

## 2. Related Work

In broad terms, the field of multi-agent RL focuses on strategic agents that interact with each other in dynamic (usually Markovian) environments. In the case of static environments, there is a large body of work on distributed cooperative optimization (Nedic & Ozdaglar, 2009; Olshevsky & Tsitsiklis, 2009) forming a solid foundation upon which several results for dynamic environments have been shown. There is also work on distributed forms of classic RL algorithms, including TD learning (Macua et al., 2014; Mathkar & Borkar, 2016).

Scalability issues for large-scale systems have been the focus of considerable work in the multi-agent RL literature (e.g., (Littman, 1994; Busoniu et al., 2008)). There are various structured sub-classes of MDPs that allow for side-stepping the computational intractability induced by the exponentially large state-action space. One broad class are factored MDPs (Kearns & Koller, 1999; Guestrin et al., 2003), in which the state space and transitions are factored, whereas the action remains a global quantity. Osband & Van Roy (2014) studies optimal policy learning for a factored MDP; their method is efficient, but requires that all agents coordinate with every other agent. There is also work on weakly coupled MDPs (Meuleau et al., 1998), in which an ensemble of MDPs with independent transitions are coupled by resource constraints.

There is also a rich literature on partially observable systems and their tractability. Optimal policies for single-agent POMDPs are typically history-dependent, computationally intractable to compute in the worst case (Papadimitriou & Tsitsiklis, 1987), and require storage scaling exponentially with the horizon. There is a large literature on various practical schemes for approximating or computing optimal policies (Smith & Simmons, 2004; Sunberg & Kochenderfer, 2018; Silver & Veness, 2010; Shani et al., 2013; Golowich et al., 2023), as well as work studying decentralized planning and learning problems for POMDPs (Oliehoek et al., 2016; Katt et al., 2018).

Early work on graph-structured or networked MDPs include the papers (Peyrard & Sabbadin, 2006; Chornei et al., 2005; Peyrard et al., 2007). In the LQR version of this problem, Shin et al. (2023) prove the existence of a near-optimal

decentralized controller. Most related to our work are some recent papers (Qu et al., 2020; Zhang et al., 2023) on local policies for graph-structured MDP models. In particular, Qu et al. (2020; 2022) propose a decentralized method for optimizing over the class of 0-hop policies, meaning that the policy depends only on the agent's local state, but do not provide guarantees on the suboptimality gap. On the other hand, Zhang et al. (2023) studies policies that are approximately optimal, but where the comparison is made to the fixed point of an entropy-regularized Bellman operator. When the regularization parameter is large, this fixed point is quite different from the standard optimal $Q$-function to which we compare.

## 3. Setup

Let $\mathcal{G}$ be a graph with $n$ nodes, and consider a graph-structured MDP with the global state space $\mathcal{S}$, global action space $\mathcal{A}$ and reward function $R : \mathcal{S} \times \mathcal{A} \to \mathbb{R}$. Using the shorthand $[n] := \{1, \dots, n\}$, we assume that the state and action space are factored, i.e., $\mathcal{S} = \otimes_{i=1}^{n} \mathcal{S}_i$ and $\mathcal{A} = \otimes_{i=1}^{n} \mathcal{A}_i$ for all $i \in [n]$, where $\mathcal{S}_i$ is the local state space and $\mathcal{A}_i$ is the local action space. In addition, we assume that the reward function is additively separable, meaning that

$$R(\boldsymbol{s}, \boldsymbol{a}) = \sum_{i=1}^{n} R_i(s_i, a_i), \qquad (1a)$$

where $R_i : \mathcal{S}_i \times \mathcal{A}_i \to \mathbb{R}$ is the local reward function of node $i$, depending only on the local state-action pair $(s_i, a_i)$.

Let $d_{\mathcal{G}}(i, j)$ denote the shortest distance on the graph between two nodes $i$ and $j$. For any integer $k = 0, 1, \dots, n-1$, we define the *$k$-hop neighborhood*

$$\mathbb{N}_i^k := \{j \in [n] \mid d_{\mathcal{G}}(j, i) \leq k\}. \qquad (1b)$$

Since node $i$ lies at distance 0, it belongs to the set $\mathbb{N}_i^k$. With this notation, we assume that the transition kernel factorizes as

$$\mathbb{P}(\boldsymbol{s}' \mid \boldsymbol{s}, \boldsymbol{a}) = \prod_{i=1}^{n} \mathbb{P}_i(s_i{}' \mid s_{\mathbb{N}_i^1}, a_{\mathbb{N}_i^1}), \qquad (1c)$$

i.e., the transition probabilities of each local state only depends on the local states and actions of its immediate neighborhood.

A *policy* $\boldsymbol{\pi}$ is a mapping from the global state space to the global action space of all the nodes, i.e., $\boldsymbol{\pi}$ is a $\mathcal{S} \to \mathcal{A}$ mapping. For a given policy $\boldsymbol{\pi}$, the $\gamma$-discounted $Q$-function is given by

$$Q^{\boldsymbol{\pi}}(\boldsymbol{s}, \boldsymbol{a}) = \mathbb{E}\Big[ \sum_{k=0}^{\infty} \gamma^k R(\boldsymbol{s}(k), \boldsymbol{a}(k)) \mid (\boldsymbol{s}(0), \boldsymbol{a}(0)) = (\boldsymbol{s}, \boldsymbol{a}) \Big],$$

where the global states and actions evolve in the following way: at any time $k$ when the whole system is at the global state $\boldsymbol{s}(k)$, the agents simultaneously choose the global action $\boldsymbol{a}(k+1) \sim \boldsymbol{\pi}(\cdot \mid \boldsymbol{s}(k))$, and the global state evolves according to the transition kernel $\boldsymbol{s}(k+1) \sim \mathbb{P}(\cdot \mid \boldsymbol{s}(k), \boldsymbol{a}(k))$. A policy $\boldsymbol{\pi}^*$ is *globally optimal* if it maximizes its $Q$-function uniformly across all state-action pairs—that is, if

$$Q^{\boldsymbol{\pi}^*}(\boldsymbol{s}, \boldsymbol{a}) \geq Q^{\boldsymbol{\pi}}(\boldsymbol{s}, \boldsymbol{a}) \quad \text{for all } (\boldsymbol{s}, \boldsymbol{a}) \text{ and } \boldsymbol{\pi}. \qquad (2)$$

Define the Bellman optimality operator $\mathscr{B}^* : \mathbb{R}^{|\mathcal{S}| \times |\mathcal{A}|} \to \mathbb{R}^{|\mathcal{S}| \times |\mathcal{A}|}$

$$\mathscr{B}^*(Q)(\boldsymbol{s}, \boldsymbol{a}) = R(\boldsymbol{s}, \boldsymbol{a}) + \gamma \sum_{\tilde{\boldsymbol{s}} \in \mathcal{S}} \mathbb{P}(\tilde{\boldsymbol{s}} \mid \boldsymbol{s}, \boldsymbol{a}) \max_{\tilde{\boldsymbol{a}} \in \mathcal{A}} Q(\tilde{\boldsymbol{s}}, \tilde{\boldsymbol{a}}). \qquad (3)$$

Standard results (Puterman, 2014) guarantee the existence of such a globally optimal policy. Since the operator $\mathscr{B}^*$ is a $\gamma$-contraction with respect to the $\ell_\infty$-norm, it has a unique fixed point $Q^*$, and this fixed point corresponds to the $Q$-function of any globally optimal policy. The globally optimal policy generally requires coordination between all the agents and full observability of the entire state space to implement. Moreover, the space and sample complexity of computing such a globally optimal policy scales with the size of the entire state-action space, rendering it impractical to implement.

## 4. Main Results

We now turn to the statements of our main results. To first build intuition, we begin in Section 4.1 by analyzing the case of a single agent. Here our main result (Theorem 1) shows how the approximate optimality of an $m$-hop policy can be controlled by a pairwise interaction matrix $\mathbf{C}$, (cf. equation (4a)); in particular, the spectral radius $\rho(\mathbf{C})$ controls the rate of decay. Building from this analysis, Section 4.3 addresses the general setting of multiple agents, and our main result (Theorem 2) gives a performance guarantee for $m$-hop co-operative policies. For the sake of brevity, we present a slightly weaker form of our error bounds; see the supplement for tighter results. Finally, in Section 4.2, we discuss the sole assumption required to prove Theorems 1 and 2 and what it means for the underlying graph-structured MDP.

### 4.1. The case of the single active agent

In the simplest case, we have a structured MDP with only a single active agent. Without loss of generality (re-indexing as needed), we assume that agent 1 is active, with a non-trivial reward and action space, and that $R_j = \boldsymbol{0}$ and $|\mathcal{A}_j| = 1$ for all agents $j \in [n] \backslash \{1\} \equiv \{2, \dots, n\}$. For a given integer $m \in [n]$, we are interested in finding an $m$-hop local

policy for agent 1, meaning a policy that depends only on states in the $m$-hop neighborhood around node 1.

A central object in our analysis is the *pairwise influence matrix* $\mathbf{C} \in \mathbb{R}_+^{n \times n}$ with $(i,j)^{th}$ entry given by

$$\mathbf{C}_{i,j} := \max_{\substack{(s,a)_j, s_{-j}, \\ (\tilde{s}, \tilde{a})_j, a_{-j}}} d_{\mathrm{TV}}(\mathbb{P}_i(\cdot | \boldsymbol{s}, \boldsymbol{a}), \mathbb{P}_i(\cdot | (\tilde{s}, \tilde{a})_j, (s,a)_{-j})),$$

(4a)

where $d_{\mathrm{TV}}$ denotes the total variation (TV) distance between two distributions.[1] The matrix entry $\mathbf{C}_{i,j}$ measures the influence of changing the local state and action of agent $j$ on the transition kernel of node $i$. By definition of the TV distance, we always have the inclusion $\mathbf{C}_{i,j} \in [0,1]$, and moreover, if agents $i, j$ are not neighbors on the graph $\mathcal{G}$, then $\mathbf{C}_{i,j} = 0$. Therefore, we can understand $\mathbf{C}$ as a type of weighted adjacency matrix. In addition, our result depends on the $k$-*hop shell* around node $i$, given by

$$\mathbb{T}_i^k := \{j \in [n] \, | \, d_{\mathcal{G}}(j,i) = k\}.$$

(4b)

**Theorem 1.** *Consider a $\gamma$-discounted graph-structured MDP with pairwise influence matrix $\mathbf{C}$ satisfying the spectral radius bound $\gamma\rho(\mathbf{C}) < 1$. Then there exists an $m$-hop policy $\tilde{\boldsymbol{\pi}}^{(m)} : \mathcal{S}_{\mathbb{N}_1^m} \to \mathcal{A}_1$ with sup-norm error $\|Q^{\tilde{\boldsymbol{\pi}}^{(m)}} - Q^*\|_\infty$ at most*

$$\frac{2\mathrm{osc}(R_1)\|\mathbf{C}\|_\infty}{1-\gamma} \sum_{\ell \in \mathbb{T}_1^m} ((\gamma\mathbf{C})^m(\mathbf{I} - \gamma\mathbf{C})^{-1})_{(1,\ell)}, \quad (5)$$

*where* $\mathrm{osc}(R_1) := \max\limits_{s_1,a_1} R_1(s_1,a_1) - \min\limits_{s_1,a_1} R_1(s_1,a_1)$.

See Section A for the proof.

Theorem 1 establishes the existence of an approximately optimal policy that depends only on the $m$-hop neighborhood of agent 1. Our analysis also shows that this policy can be computed in an efficient way. In particular:

- Given knowledge of the state transition kernels for each state in $\mathcal{S}_{\mathbb{N}_1^m}$, the policy $\tilde{\boldsymbol{\pi}}^{(m)}$ can be computed in polynomial time with the space complexity $|\mathcal{S}_{\mathbb{N}_1^m}||\mathcal{A}_{\mathbb{N}_1^m}|$.

- When state transitions are not known, the policy $\tilde{\boldsymbol{\pi}}^{(m)}$ can be approximated from samples using a fitted $Q$-iteration (FQI) over the $m$-neighborhood.

The key component of the performance bound (5) is the the term $((\gamma\mathbf{C})^m(1 - \gamma\mathbf{C})^{-1})_{(1,\ell)}$: it characterizes the long-range effect of ignoring the local states for each node $\ell$ outside the $m$-ball surrounding node 1. Given our bound on

---

[1]For use in the sequel, we have defined the matrix $\mathbf{C}$ for the general multi-agent setting; for the single agent setting under consideration in this section, only agent 1 has a non-trivial action space.

the spectral radius, the term $(\gamma\mathbf{C})^m$ decays exponentially in $m$, and reflects the fact that any state changes outside the $m$-neighborhood require $m$ steps to impact the reward of agent 1.

As shown in the proof, Theorem 1 can be strengthened by introducing the $m$-*hop influence matrix* with entries

$$\mathbf{H}_{i,j}^{(m)} := \max_{\substack{\boldsymbol{s}, \tilde{\boldsymbol{s}} \\ \boldsymbol{a}, \tilde{\boldsymbol{a}}}} d_{\mathrm{TV}}\Big(\mathbb{P}_j(\cdot \mid \boldsymbol{s}, \boldsymbol{a}),$$

$$\mathbb{P}_j(\cdot \mid (s,a)_{\mathbb{N}_i^m}, (\tilde{s}, \tilde{a})_{(\mathbb{N}_i^m)^c})\Big). \quad (6a)$$

This quantity tracks the effect of changing the local states and actions outside the $m$-hop neighborhood of agent $i$ on the state of node $j$. Note that $\mathbf{H}_{i,j}^{(m)} = 0$ for any $j$ such that $d_{\mathcal{G}}(i,j) < m$. With this definition, we show in the proof that the performance gap $\|Q^{\tilde{\boldsymbol{\pi}}^{(m)}} - Q^*\|_\infty$ is at most

$$\underbrace{\frac{2}{1-\gamma} \mathrm{osc}(R_1) \sum_{\ell \in \mathbb{T}_1^m} \mathbf{H}_{1,\ell}^{(m)}((\gamma\mathbf{C})^m(1 - \gamma\mathbf{C})^{-1})_{(1,\ell)}}_{\equiv U^{(m)}}$$

(6b)

which is a refinement of the bound (5).

**Proof sketch:** So as to provide intuition, let us describe some key ideas from the proof; see Section A for the technical argument itself. Theorem 1 guarantees the existence of an $m$-hop policy $\tilde{\boldsymbol{\pi}}^{(m)}$ with the prescribed optimality guarantees; the proof itself is explicit and constructive in nature. For any hopsize $m = 0, 1, \ldots$, we define the projection operator

$$(\Pi^{(m)} Q)(\boldsymbol{s}, a_1) := Q(s_{\mathbb{N}_1^m}, \overline{s}_{n \setminus \mathbb{N}_1^m}, a_1), \quad (7)$$

which fixes all states outside the $m$-hop neighborhood to some fixed state $\overline{s}$. By construction, the operator $\Pi^{(m)}$ is non-expansive with respect to the $\ell_\infty$-norm. As a result, the projected Bellman operator $\Pi^{(m)} \circ \mathscr{B}^*$ is $\gamma$-contractive in the $\ell_\infty$-norm. By the contraction mapping principle, there is a unique $Q^\dagger$ satisfying the fixed point relation $\Pi^{(m)}(\mathscr{B}^*(Q^\dagger)) = Q^\dagger$. Our analysis applies to any policy $\tilde{\boldsymbol{\pi}}^{(m)}$ that is greedy with respect to $Q^\dagger$; since the projected fixed point $Q^\dagger$ depends only on variables on the $m$-hop neighborhood, any such greedy policy is $m$-hop local by definition. Given knowledge of the reward and transition dynamics, the function $Q^\dagger$ can be computed by the standard fixed point iteration $Q_{t+1}^{(m)} = \Pi^{(m)}(\mathscr{B}^*(Q_t^{(m)}))$. By the $\gamma$-contractivity, the iterates $Q_t^{(m)}$ converge geometrically fast to $Q^\dagger$.

The bulk of our technical effort is devoting to upper bounding the Bellman residual sup-norm error $\|\mathscr{B}^*(Q^\dagger) - Q^\dagger\|_\infty$ of the projected fixed point, and we do so by first bounding

the difference

$$\Delta_{t+1}^{(m)}(\boldsymbol{s}, a_1) := \mathscr{B}^* Q_t^{(m)}(\boldsymbol{s}, a_1) - Q_{t+1}^{(m)}(\boldsymbol{s}, a_1)$$

along the iterates of the algorithm. The key technical step establishes the uniform upper bound

$$\Delta_{t+1}^{(m)}(\boldsymbol{s}, a_1) \leq \mathrm{osc}(R_1) \sum_{\ell \in \mathbb{T}_1^m} \mathbf{H}_{1,\ell}^{(m)} \Big[ \gamma^m \mathbf{C}^m (\mathbf{I} - \gamma \mathbf{C})^{-1} \Big]_{(1,\ell)},$$

Observe that the right-hand side of this bound is independent of the iteration number $t$, so that we can take the limit as $t \to \infty$. Doing so leads to a bound on the Bellman residual of the projected fixed point $Q^\dagger$, as in equation (6b).

### 4.2. Spectral Radius and Graph Topology

Let us discuss the spectral radius bound $\gamma \rho(\mathbf{C}) < 1$ under which Theorem 1 is applicable. From its definition (4a), the pairwise influence matrix $\mathbf{C}$ is graph-structured, with non-zeros only for node pairs $(i, j)$ that are connected in the graph; for any such edge, the associated weight $\mathbf{C}_{i,j}$ also depends on the transition structure. In the sequel (Section 5), we give numerical results that illustrate how both the graph topology and interaction strengths affect the decay rates.

The spectral radius can be upper bounded by any matrix norm, so that (for instance) it satisfies the upper bounds

$$\rho(\mathbf{C}) \leq \min \Big\{ \underbrace{\max_{j \in [n]} \sum_{i=1}^{n} |\mathbf{C}_{i,j}|}_{\equiv \|\mathbf{C}\|_1}, \underbrace{\max_{i \in [n]} \sum_{j=1}^{n} |\mathbf{C}_{i,j}|}_{\equiv \|\mathbf{C}\|_\infty} \Big\}. \quad (8a)$$

Since $\mathbf{C}$ is not symmetric in general, both upper bounds are useful. More crudely, the inequality (8a) implies the (very conservative) upper bound $\rho(\mathbf{C}) \leq d_{\max} \|\mathbf{C}\|_{\max}$, where $d_{\max}$ is the maximum degree of the graph and $\|\mathbf{C}\|_{\max}$ is the maximum entry of $\mathbf{C}$. (Recall that by definition of the TV norm, we always have the upper bound $\|\mathbf{C}\|_{\max} \leq 1$.) Thus, the spectral radius can be controlled by the maximum degree of the graph $\mathcal{G}$ and the maximum interaction strength between any two agents.

We can obtain a more refined dependence by exploiting bounds from spectral graph theory. Let $\mathbf{A}$ be the adjacency matrix of the graph $\mathcal{G}$, we always have the entrywise upper bound

$$\mathbf{C} \preceq_{\text{e.w.}} \|\mathbf{C}\|_{\max} \mathbf{A}. \quad (8b)$$

Since both matrices have non-negative entries, the Perron-Frobenius theorem implies that $\rho(\mathbf{C}) \leq \|\mathbf{C}\|_{\max} \rho(\mathbf{A})$. Therefore, the structure of the underlying graph directly affects $\rho(\mathbf{C})$, which controls the rate of exponential decay in all of our suboptimality gaps. The effect of the spectral radius of the adjacency matrix has been noted before in

various network problems (e.g., (Van Mieghem et al., 2008; Duchi et al., 2012)).

As noted previously, we always have the worst-case bound $\rho(\mathbf{C}) \leq d_{\max}$ where $d_{\max}$ is the maximum degree, and this bound is achieved with equality by a $d_{\max}$-regular graph (meaning that each node has exactly degree $d_{\max}$). For graphs without cycles (known as trees; see panel (a) in Figure 1 for an example), the spectral radius admits the sharper upper bound $\rho(\mathbf{C}) \leq 2\sqrt{d_{\max} - 1}$. Thus, our theory suggests that that tree graphs have a faster decay in $m$ that $d_{\max}$-regular graphs, and this effect is present in our simulations (compare the tree in column (a) to the grid in column (c) in Figure 1).

### 4.3. The multi-agent problem

In Section 4.1, we established the existence of approximately-optimal policies that only use information from an $m$-hop neighborhood in the case of a single active agent. Here we extend these results to the multi-agent case. The broad conclusion remains the same: for a given global state configuration, each agent can compute their optimal action in terms of $Q$-functions that depend only on their $m$-hop neighborhood, but with the subtlety that some form of coordination can be required in choosing optimal actions.

We define an $m$-*hop cooperative policy* $\tilde{\boldsymbol{\pi}}^{(m)} : \mathcal{S} \to \mathcal{A}$ to be a policy that is greedy with respect to the sum of $Q$-functions, each of which only depends on an $m/2$-hop neighborhood. In analytical terms, we write

$$\tilde{\boldsymbol{\pi}}^{(m)}(\boldsymbol{s}) \in \arg\max_{\boldsymbol{a'}} \sum_i \tilde{Q}_i((s, a')_{\mathbb{N}_i^{m/2}}), \quad (9)$$

for some set of $Q$-functions $\{Q_i\}_{i \in [n]}$, where each $Q_i$ is a function only of agent $i$'s $m/2$-hop neighborhood.

While the action of any agent $j$ does depend on the global state $\boldsymbol{s}$, there is additional local structure here: each agent only needs to communicate with its $m$-hop neighborhood to find what its own action to be played is. To see this fact more clearly, let $\tilde{Q}(\boldsymbol{s}, \boldsymbol{a}) := \sum_i \tilde{Q}_i((s, a')_{\mathbb{N}_i^{m/2}})$ for brevity. Then the optimality gap of action $a_j$ compared to $a'_j$ equals

$$\tilde{Q}(\boldsymbol{s}, \boldsymbol{a}) - \tilde{Q}(s_j, a'_j, (s, a)_{-j}) = \sum_{i \in \mathbb{N}_j^{m/2}} \big( \tilde{Q}_i((s, a)_{\mathbb{N}_i^{m/2}})$$

$$- \tilde{Q}_i(s_j, a'_j, (s, a)_{\mathbb{N}_i^{m/2} \setminus \{j\}}) \big)$$

Note that the right-hand side depends on the states and actions in the $m$-hop neighborhood of agent $j$. This is because each $\tilde{Q}_i$ for $i \in \mathbb{N}_j^{m/2}$ depends on agent $i$'s $m/2$ neighborhood, and hence giving an overall dependence on agent $i$'s $m$-hop neighborhood. If agent $i$ were to receive the $Q_j$'s in its $m/2$-hop neighborhood and know the states and actions of everyone in its $m$-hop neighborhood, it would be

able to compute what its own action is to follow the $m$-hop cooperative policy. There are also several algorithms that aim to solve optimization problems like in equation (9) in a decentralized manner, such as consensus and gossip algorithms (Shah, 2009), and the message-passing algorithms on graphs (Kschischang et al., 2002; Wainwright & Jordan, 2008). The class of $m$-hop cooperative policies encode various ways of cooperation between agents within their $m$-hop neighborhoods.

For a central planner who has access to the specifications of the entire system, the space complexity of storing and implementing these $m$-hop cooperative policies is only $\sum_i |\mathcal{S}_{\mathbb{N}_i^m}||\mathcal{A}_{\mathbb{N}_i^m}|$, which is at best linear in the number of agents $n$. This is as opposed to the space complexity $|\mathcal{S}||\mathcal{A}|$ of storing a general globally optimal policy that can scale, in the worst case, exponentially in $n$. Note that this is a natural generalization of the $m$-hop policy defined in the single-agent case in Theorem 1.

### 4.4. Existence result

Our main result for the multi-agent case shows the existence of an approximately-optimal $m$-hop cooperative policy with the error decaying exponentially with $m$. In addition to the pairwise influence matrix $\mathbf{C}$, it involves a second type of influence matrix. Let $\mathbf{W}$ be the two-hop adjacency matrix of the graph with entries

$$[\mathbf{W}]_{i,j} := \mathbb{I}[d_{\mathcal{G}}(j,i) \leq 2] = \begin{cases} 1 & \text{if } d_{\mathcal{G}}(j,i) \leq 2 \\ 0 & \text{otherwise,} \end{cases} \quad (10a)$$

and consider the discrete-time Lyapunov equation

$$\mathbf{D} = \mathbf{W} + \gamma \mathbf{C}^\top \mathbf{D} \mathbf{C}. \quad (10b)$$

Under the spectral radius bound $\sqrt{\gamma}\rho(\mathbf{C}) < 1$, standard results (Horn & Johnson, 2012) ensure that this equation has a unique solution $\mathbf{D}$. Our final bound involves the matrix $\mathbf{M} \in \mathbb{R}^{n \times n}$ with entries

$$\mathbf{M}_{i,\ell} := ((\sqrt{\gamma}\mathbf{C}^\top)^{(m/4-2)}\mathbf{D}(\sqrt{\gamma}\mathbf{C})^{(m/4-2)})_{(i,\ell)}. \quad (10c)$$

**Theorem 2.** *Consider a $\gamma$-discounted graph-structured MDP with pairwise influence matrix $\mathbf{C}$ satisfying the spectral radius bound $\sqrt{\gamma}\rho(\mathbf{C}) < 1$. Then for any $m \geq 8$, there exists an $m$-hop cooperative policy $\tilde{\boldsymbol{\pi}}^{(m)}$ with suboptimality gap $\|Q^{\tilde{\boldsymbol{\pi}}^{(m)}} - Q^*\|_\infty$ uniformly upper bounded by*

$$\frac{4\mathrm{osc}(R)\gamma^2\|\mathbf{C}\|_\infty}{(1-\gamma)^2} \sum_{i \in [n]} \sum_{\ell \in \left(\mathbb{N}_i^{m/2-1}\right)^c} \mathbf{M}_{i,\ell} \quad (11)$$

See Section B for the proof.

Comparing with Theorem 1, the error bound has a similar structure. For each agent $i$, there is a sum over the nodes outside the $m/2$-hop ball around it. For each agent $\ell$ outside the $m/2$-hop ball, the decay term $\mathbf{M}_{i,\ell}$ characterizes the long-range effect of ignoring information outside the $m/2$-hop ball.

The main crux of the proof is in showing that the optimal $Q$-function $Q^*$ can be approximated by a sum of such $Q$-functions that only depend on $m$-hop neighborhoods. The error terms in Theorem 2 that arose due to ignoring information outside the $m/2$-hop ball is precisely this approximation error in capturing $Q^*$.

As was the case in Theorem 1, the bound on the performance gap $\|Q^{\tilde{\boldsymbol{\pi}}^{(m)}} - Q^*\|_\infty$ can be improved using the $m$-hop influence matrix (6a). In particular, in the notation of Theorem 2, we have the upper bound

$$\frac{4\mathrm{osc}(R)\gamma^2\|\mathbf{C}\|_\infty}{(1-\gamma)^2} \sum_{i \in [n]} \sum_{\ell \in \left(\mathbb{N}_i^{m/2-1}\right)^c} \mathbf{H}_{i,\ell}^{(m)}\mathbf{M}_{i,\ell}.$$

**Proof sketch:** Here we provide some high-level comments on the proof of Theorem 2; see Section B for the full details. The proof of this result is more involved than the single-agent guarantee in Theorem 1. In the former case, we were able to directly analyze the dynamics of the projected $Q$-updates, and thereby bound the Bellman residual of the projected fixed point $Q^\dagger$.

In the multi-agent case, a direct generalization of the projected $Q$-iteration does not work. Naively fixing state-actions outside agent $i$'s $m$-hop neighborhood can lead to large projection errors, due to the possibility of agents lying outside the $m$-hop neighborhood that nonetheless have a strong effect on the optimal $Q$-function. We circumvent this challenge by instead constructing a surrogate function $\widetilde{Q}^{(m)}$ that consists of a sum of projections onto $m/2$-sized neighborhoods. Our $m$-cooperative policy is then greedy with respect to this decomposition. However, the operations that define the surrogate $\widetilde{Q}^{(m)}$ do *not* lead to a non-expansive operator, so that we cannot exploit a contraction analysis. Controlling the additional errors via different techniques leads to the extra factor of $(1-\gamma)$ in the denominator of the bound Equation (11).

Moreover, while the proof of Theorem 2 is constructive, in the sense that we exhibit a particular $m$-surrogate $\widetilde{Q}^{(m)}$, it is currently defined directly in terms of direct operations on the optimal function $Q^*$. It remains an open problem to determine an efficient algorithm for computing the surrogate $\widetilde{Q}^{(m)}$, or a related quantity with similar guarantees.

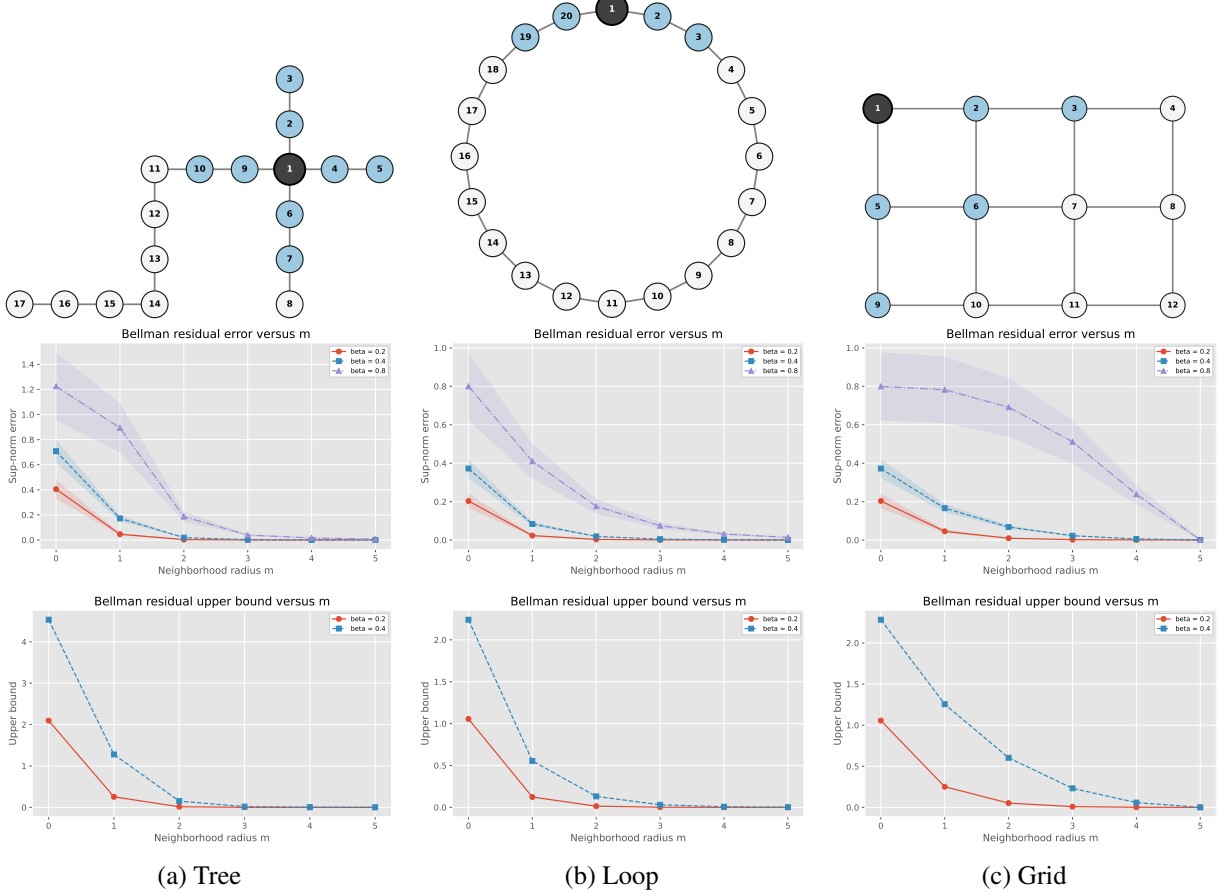

**Figure 1.** Behavior of $m$-hop local policies for three different graph types, and a simple binary infection model with the dynamics (12). Top row: illustrations of three different graphs, with active node 1 marked in black; the neighborhood set for hopsize $m = 2$ marked in blue; and remaining nodes in white. Middle row: Plots of the sup-norm Bellman residual error versus the $m$-hop policy size. Each curve corresponds to a different choice of connection strength $\beta$ from the transition dynamics (12). Each plot shows the mean over ten trials with randomly constructed reward types, and shaded areas correspond to the standard errors over the reward randomness. Bottom row: Plots of the upper bound $U^{(m)}$ from equation (6b) versus the hopsize $m$. These plots are limited to the interaction pair $\beta \in \{0.2, 0.4\}$.

## 5. Numerical Results

In broad terms, the main take-aways from our theory are the existence of $m$-hop policies whose sub-optimality gap decays exponentially in $m$. In this section, we report the results of some simple numerical results that illustrate the form of these predictions for different graph topologies.

**Binary infection model:** Given a graph with $n$ nodes, we considered a binary infection model with state space $\mathcal{S}_i = \{-1, +1\}$ at each node $i \in [n]$, with $s_i = +1$ corresponding to the infected state and $s_i = -1$ to the healthy state. Thus, the full state space $\boldsymbol{\mathcal{S}} = \otimes_{i=1}^n \mathcal{S}_i$ has cardinality $2^n$. Node 1 is associated with a binary action $a_1 \in \{0, 1\}$, where $a_1 = 1$ corresponds to a "treat" decision (e.g., a drug); all other nodes $j \neq 1$ have only $a_j = 0$. Letting $\sigma(t) = \frac{e^t}{1+e^t}$ denote the logistic function, the transition

function $\mathbb{P}_i(s_i = +1 \mid s_{\mathbb{N}_i^1}, a_i)$ takes the form

$$\sigma\Big(b + \rho s_i + \beta \sum_{j \in \mathbb{N}_i^1 \setminus \{i\}} s_j - \alpha a_i \mathbb{I}[s_i = 1]\Big). \quad (12)$$

Here $b \in \mathbb{R}$ represents *base susceptibility* of individual $i$ to the infection, with negative values biasing towards the uninfected state $s_i = -1$. The parameter $\rho \geq 0$ represents *individual infection persistence*, controlling how strongly the current state $s_i$, whether healthy or infected, propagates forward in time. The $\beta \geq 0$ represents the strength of *neighborhood infection propagation*, with larger values meaning greater probability of infection when many of node $i$'s neighbors are infected. Finally, the *treatment effect parameter* $\alpha > 0$ controls the strength of taking action $a_1 = 1$ to vaccinate. Note that $\alpha$ is irrelevant for any node $j \neq 1$, since in this case we have $a_j = 0$ always.

In the simulations reported here, we implemented these

dynamics with $b = -0.2$, $\rho = 0.6$, $\alpha = 1.0$, and the neighborhood strength $\beta$ varying in the set $\{0.2, 0.4, 0.8\}$. In all cases, we instantiated agent rewards of the form

$$
R_1(s_1, a_1) = \begin{array}{c} \\ s_1 = -1 \\ s_1 = +1 \end{array} \overset{\begin{array}{cc} a_1 = 0 & a_1 = 1 \end{array}}{\left[ \begin{array}{cc} 2.0 & 1.8 \\ -2.0 & -0.4 \end{array} \right]},
$$

so that the treatment action $a_1 = 1$ has a strong positive effect (from $-2.0$ to $-0.4$) when in an infected state $s_1 = 1$, and a mild negative effect (from $2.0$ to $1.8$) when in a healthy state $s_1 = -1$. In order to generate ensembles, we perturbed the size of each effect size by a random uniform perturbation $\xi \in \mathrm{Uni}[0, 0.5]$.

We performed simulations for three different graph topologies, as illustrated in the top row of Figure 1. For each graph, choice of $\beta$, reward perturbation, and neighborhood radius $m$, we computed the fixed point $Q_m^\dagger$ of the projected Bellman iteration $Q_{t+1}^{(m)} = \Pi^{(m)} \mathscr{B}^* Q_t^{(m)}$ up to accuracy $\epsilon = 0.001$. We then evaluated the theorem-relevant Bellman residual $\|\mathscr{B}^*(Q_m^\dagger) - Q_m^\dagger\|_\infty$ using the full graph transition kernel in the Bellman operator (3). For each graph and $(\beta, m)$-pair, we repeated this computation for $10$ random perturbations of the reward function, and then computed the average sup-norm Bellman residual over these trials, along with 95% CIs over the $10$ reward perturbations (to assess fluctuations with reward settings).

The panels in the middle row of Figure 1 give plots of these sup-norm residuals versus the neighborhood hopsize $m$. Each plot corresponds to a different graph topology, with the three curves corresponding to the interaction strengths $\beta \in \{0.2, 0.4, 0.8\}$. Consistent with Theorem 1, each of these curves exhibit rapid decay as the hopsize $m$ is increased, with slower decay rates for larger values of $\beta$. The graph topology also plays a role, with the more densely connected grid graph exhibiting the slowest rates of convergence, especially for larger values of $\beta$. (For the grid graph, overly large values of $\beta$ no longer lead to rapid decay, with the spectral radius condition being violated eventually.)

Finally, we numerically evaluated the upper bounds predicted by our theory; in particular, so as to compare directly to Bellman residuals, we compute the term $U^{(m)}$ defined in the refined upper bound (6b). The panels in the bottow row of Figure 1 give plots of this upper bound $U^{(m)}$ versus the hopsize $m$. (We limited ourselves to plots for the interaction strengths $\beta \in \{0.2, 0.4\}$, since the spectral radius bound was not satisfied by all graphs for the setting $\beta = 0.8$.) In numerical terms, these bounds are conservative by a factor of 4 to 5 for the smaller hopsizes $m \in \{0, 1\}$, but exhibit the exponential decay guaranteed by our theory, and are qualitatively consistent with actual Bellman residuals in the middle row.

## 6. Discussion

In this paper, we studied the near-optimality of local policies for graph-structured MDPs, in particular those that depend only on the states in an $m$-hop neighborhood around a given agent. Our main results involve pairwise influence matrices that reflect the underlying graph topology, and capture the interaction strength between any two agents. We proved two main results that characterize the sub-optimality of $m$-local policies. For an influence matrix with suitably bounded spectral radius, we provide upper bounds on the sub-optimality that decay exponentially in the hopsize $m$. In the multi-agent setting, we propose approximately-optimal policies that are defined by $m$-local functions, but require agents to communicate with their neighbors and coordinate their actions. This is a rich class of policies that, to our knowledge, has not been studied before in the literature.

Our work opens multiple avenues for future study. In the single agent setting, we showed how to compute an approximately-optimal $m$-hop policy in space- and time-efficient manner. Results of such calculations were illustrated via a numerical study on an epidemic model in Section 5. In the more challenging multi-agent setting, we established the existence of an $m$-cooperative policy with bounds on sub-optimality, but it remains to give a computationally efficient procedure for computing such a greedy policy. Finally, our analysis focused on the cooperative setting, in which all agents share the goal of maximizing a separable reward function over time. It would also be interesting to see if similar results hold for a competitive setting. Specifically, are there any notions of equilibria that can be approximated when each agent uses only local information?

## Impact statement

This paper presents work whose goal is to advance the field of Machine Learning. There are many potential societal consequences of our work, none that we feel must be specifically highlighted here.

### Acknowledgements

This work was partially funded by ONR Grant N00014026-1-2116 and NSF DMS-2311072 for MJW. FZF was supported by the Mathworks fellowship.

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

# A. Proof of Theorem 1

We prove the stronger bound (6b) that involves the $m$-hop influence matrix $\mathbf{H}^{(m)}$ from equation (6a).

Our analysis exploits the following standard result (cf. Prop 2.2.1 in Bertsekas (2022)). Let $\boldsymbol{\pi}$ be a greedy policy induced by some function $Q$, and let $Q^{\boldsymbol{\pi}}$ be the $Q$-value function of the policy $\boldsymbol{\pi}$. Then we have the uniform bound

$$\|Q^{\boldsymbol{\pi}} - Q^*\|_\infty \leq \frac{2}{1-\gamma}\|\mathscr{B}^*(Q) - Q\|_\infty, \tag{13}$$

where $\mathscr{B}^*$ is the Bellman optimality operator (3). Thus, the Bellman residual $\mathscr{B}^*(Q) - Q$ can be used to control the sub-optimality of the greedy policy induced by $Q$.

Given the bound (13), we have reduced the problem to controlling the Bellman residual of a well-chosen $Q$-function. In particular, we consider the fixed point of a projected Bellman operator. For any hopsize $m = 0, 1, \ldots$, we define the projection operator

$$(\Pi^{(m)}Q)(\boldsymbol{s}, a_1) := Q(s_{\mathbb{N}_1^m}, \overline{s}_{n\backslash\mathbb{N}_1^m}, a_1),$$

which fixes all states outside the $m$-hop neighborhood to some fixed state $\overline{\boldsymbol{s}}$. Observe that $\Pi^{(m)}$ is a non-expansive operator with respect to the $\ell_\infty$-norm. As a result, the projected Bellman operator $\Pi^{(m)} \circ \mathscr{B}^*$ is $\gamma$-contractive in the $\ell_\infty$-norm. The contraction mapping principle implies there is a unique $Q^\dagger$ satisfying the fixed point relation $\Pi^{(m)}(\mathscr{B}^*(Q^\dagger)) = Q^\dagger$, and moreover, that the iterates $Q_{t+1}^{(m)} = \Pi^{(m)}\mathscr{B}^* Q_t^{(m)}$ converge to $Q^\dagger$ from any initialization.

In order to complete the proof, it suffices to show that

$$\|\mathscr{B}^*(Q^\dagger) - Q^\dagger\|_\infty \leq \mathrm{osc}(R_1) \sum_{\ell \in \mathbb{T}_1^m} \mathbf{H}_{1,\ell}^{(m)} \left[\gamma^m \mathbf{C}^m (\mathbf{I} - \gamma\mathbf{C})^{-1}\right]_{(1,\ell)}. \tag{14a}$$

Our first step is to bound the differences $\Delta_{t+1}^{(m)}(\boldsymbol{s}, a_1) := \mathscr{B}^* Q_t^{(m)}(\boldsymbol{s}, a_1) - Q_{t+1}^{(m)}(\boldsymbol{s}, a_1)$ along the iterates of the algorithm. In particular, we show that

$$\Delta_{t+1}^{(m)}(\boldsymbol{s}, a_1) \leq \mathrm{osc}(R_1) \sum_{\ell \in \mathbb{T}_1^m} \mathbf{H}_{1,\ell}^{(m)} \left[\gamma^m \mathbf{C}^m (\mathbf{I} - \gamma\mathbf{C})^{-1}\right]_{(1,\ell)}, \tag{14b}$$

Note that the right-hand side of the bound (14b) is independent of the iteration number $t$, as well as the choice of $(\boldsymbol{s}, a_1)$. Thus, the bound remains valid as we take $t \to \infty$, so that

$$[\mathscr{B}^*(Q^\dagger) - Q^\dagger](s, a) = \lim_{t \to \infty} \Delta_{t+1}^{(m)}(\boldsymbol{s}, a_1) \leq \mathrm{osc}(R_1) \sum_{\ell \in \mathbb{T}_1^m} \mathbf{H}_{1,\ell}^{(m)} \left[\gamma^m \mathbf{C}^m (\mathbf{I} - \gamma\mathbf{C})^{-1}\right]_{(1,\ell)}.$$

Taking the supremum over $(\boldsymbol{s}, a_1)$ yields the $\ell_\infty$-bound (14a).

**Proof of the bound (14b):** It remains to prove the intermediate claim (14b), which we do via two lemmas. For any $Q$-function $Q$ and any node $\ell \in [n]$, define the local difference

$$\delta_\ell(Q) := \sup_{s_{-\ell}} \sup_{s_\ell, \tilde{s}_\ell} |Q(s_{-\ell}, s_\ell, a_1) - Q(s_{-\ell}, \tilde{s}_\ell, a_1)|. \tag{15}$$

This quantity tracks the effect of changing the state of node $\ell$ on $Q$. Our first lemma bounds the Bellman residual in terms of the matrix $\mathbf{H}^{(m)}$ and the local difference terms.

**Lemma 1.** *For each iteration $t = 1, 2, \ldots$, the Bellman residual between successive iterates is upper bounded as*

$$\underbrace{\left|\left[(\mathscr{B}^* Q_t^{(m)}) - Q_{t+1}^{(m)}\right](\boldsymbol{s}, a_1)\right|}_{\equiv |\Delta_{t+1}^{(m)}(s,a)|} \overset{(i)}{\leq} \sum_{\ell \in \mathbb{N}_1^m} \mathbf{H}_{1,\ell}^{(m)} \delta_\ell(Q_t^{(m)}) \overset{(ii)}{\leq} \sum_{\ell \in \mathbb{T}_1^m} \mathbf{H}_{1,\ell}^{(m)} \delta_\ell(Q_t^{(m)}) \tag{16}$$

See Section A.1 for the proof of this claim. Equality (ii) follows since $\mathbf{H}_{1,\ell}^{(m)} = 0$ for all nodes $\ell$ such $d_{\mathcal{G}}(1,\ell) < m$, so we can restrict the sum to the shell $\mathbb{T}_1^m = \{\ell \in [n] \mid d_{\mathcal{G}}(1,\ell) = m\}$.

Our next step is to bound the local difference terms.

**Lemma 2** (Bounding $\delta_\ell(Q_t^{(m)})$). *For each iteration $t = 0, 1, 2, \ldots$ and node $\ell \in [n]$, we have*

$$\delta_\ell(Q_t^{(m)}) \leq \operatorname{osc}(R_1)\Big\{\mathbb{I}(\ell = 1) + \sum_{r=1}^{t} \gamma^r (\mathbf{C}^r)_{(1,\ell)}\Big\}, \tag{17}$$

*where $(\mathbf{C}^r)_{(1,\ell)}$ is the $(1,\ell)^{th}$ element of $\mathbf{C}^r$.*

See Section A.2 for the proof of this claim.

We now complete the proof by combining the pieces. Recall that $\mathbf{C}$ has the same zero pattern as the graph adjacency matrix. Consequently, for any $\ell \in \mathbb{T}_1^m$, we have $(\mathbf{C}^r)_{(1,\ell)} = 0$ all $r = 1, 2, \ldots, m-1$, since the shortest path between node 1 and node $\ell$ has length $m$ by definition of $\mathbb{T}_1^m$. Consequently, for any iteration $t \geq m$, the bound (17) implies that

$$\delta_\ell(Q_t^{(m)}) \leq \operatorname{osc}(R_1) \sum_{r=m}^{t} \gamma^r (\mathbf{C}^r)_{(1,\ell)} \qquad \text{for all } \ell \in \mathbb{T}_1^m.$$

Combining this upper bound with inequality (16) yields

$$\begin{aligned}
\frac{\Delta_{t+1}^{(m)}(s,a)}{\operatorname{osc}(R_1)} &\leq \sum_{\ell \in \mathbb{T}_1^m} \mathbf{H}_{1,\ell}^{(m)} \sum_{r=m}^{t} \gamma^r (\mathbf{C}^r)_{(1,\ell)} = \sum_{\ell \in \mathbb{T}_1^m} \mathbf{H}_{1,\ell}^{(m)} \Big[\gamma^m \mathbf{C}^m \sum_{r=0}^{t-m} \gamma^r \mathbf{C}^r\Big]_{(1,\ell)} \\
&\overset{(iii)}{\leq} \sum_{\ell \in \mathbb{T}_1^m} \mathbf{H}_{1,\ell}^{(m)} \Big[\gamma^m \mathbf{C}^m \sum_{r=0}^{\infty} \gamma^r \mathbf{C}^r\Big]_{(1,\ell)} \\
&\overset{(iv)}{=} \sum_{\ell \in \mathbb{T}_1^m} \mathbf{H}_{1,\ell}^{(m)} \Big[\gamma^m \mathbf{C}^m (\mathbf{I} - \gamma \mathbf{C})^{-1}\Big]_{(1,\ell)},
\end{aligned}$$

where inequality (iii) follows since $\mathbf{C}$ has non-negative entries, and equality (iv) follows since the assumption $\gamma \rho(\mathbf{C}) < 1$ ensures that the geometric sum converges. This completes the proof of the bound (14b).

**Recovering the weaker bound** (5)**:** In order to recover the weaker bound (5) stated in Theorem 1, we need to show that $\mathbf{H}_{1,\ell}^{(m)} \leq \|\mathbf{C}\|_\infty$.

Consider the set $\mathcal{I}^m := [n] \setminus \mathbb{N}_1^m$ of nodes that lie outside agent 1's $m$-hop neighborhood. We have

$$\begin{aligned}
\mathbf{H}_{1,\ell}^{(m)} &= \max_{\substack{s,\tilde{s} \\ a,\tilde{a}}} d_{\mathrm{TV}}\Big(\mathbb{P}_\ell(\cdot \mid s,a), \mathbb{P}_\ell(\cdot \mid (s,a)_{\mathbb{N}_1^m}, (\tilde{s},\tilde{a})_{[n]\setminus\mathbb{N}_1^m})\Big) \\
&\overset{(i)}{\leq} \max_{\substack{s,\tilde{s} \\ a,\tilde{a}}} \sum_{j \in I_{-i}^m} d_{\mathrm{TV}}\Big(\mathbb{P}_\ell(\cdot \mid (s,a)_{\mathbb{N}_1^m}, (\tilde{s},\tilde{a})_{v \in I_{-1}^m, v<j}, (s,a)_{v \in I_{-1}^m, v \geq j}), \mathbb{P}_\ell(\cdot \mid (s,a)_{\mathbb{N}_1^m}, (\tilde{s},\tilde{a})_{v \in I_{-1}^m, v \leq j}, (s,a)_{v \in I_{-1}^m, v>j})\Big) \\
&\overset{(ii)}{\leq} \sum_{j \in \mathcal{I}^m} \mathbf{C}_{1,j} \\
&\leq \max_{k=1,\ldots,[n]} \sum_{j=1}^{n} \mathbf{C}_{k,j} \overset{(iii)}{=} \|\mathbf{C}\|_\infty,
\end{aligned}$$

where step (i) follows from the triangle inequality; step (ii) follows since each term in the sum depends only on the pairwise interaction between agents $\ell$ and $v$, which is controlled via $\mathbf{C}$; and step (iii) follows by the definition the induced $\ell_\infty$-matrix operator norm.

## A.1. Proof of Lemma 1

Using the update equation for $Q_{t+1}^{(m)}$, we can write

$$
\begin{aligned}
(\mathscr{B}^* Q_t^{(m)})(\boldsymbol{s}, a_1) - Q_{t+1}^{(m)}(\boldsymbol{s}, a_1) &= (\mathscr{B}^* Q_t^{(m)})(\boldsymbol{s}, a_1) - (\Pi^{(m)} \mathscr{B}^* Q_t^{(m)})(\boldsymbol{s}, a_1) \\
&= (\mathscr{B}^* Q_t^{(m)})(\boldsymbol{s}, a_1) - (\mathscr{B}^* Q_t^{(m)})(s_{\mathbb{N}_1^m}, \tilde{s}_{\mathbb{N}_1^{-m}}, a_1) \\
&\overset{(i)}{=} \sum_{\boldsymbol{s}'} (\mathbb{P}(\boldsymbol{s}' \mid \boldsymbol{s}, a_1) - \mathbb{P}(\boldsymbol{s}' \mid s_{\mathbb{N}_1^m}, \tilde{s}_{\mathbb{N}_1^{-m}}, a_1)) \max_{\tilde{a}_1} Q_t^{(m)}(\boldsymbol{s}', \tilde{a}_1) \\
&\overset{(ii)}{=} \sum_{\boldsymbol{s}'} (\mathbb{P}(\boldsymbol{s}' \mid \boldsymbol{s}, a_1) - \mathbb{P}(\boldsymbol{s}' \mid s_{\mathbb{N}_1^m}, \tilde{s}_{\mathbb{N}_1^{-m}}, a_1)) \max_{\tilde{a}_1} Q_t^{(m)}(s'_{\mathbb{N}_1^m}, \tilde{a}_1) \\
&\overset{(iii)}{=} \sum_{s'_{\mathbb{N}_1^m}} (\mathbb{P}(s'_{\mathbb{N}_1^m} \mid s_{\mathbb{N}_1^{m+1}}, a_1) - \mathbb{P}(s'_{\mathbb{N}_1^m} \mid s_{\mathbb{N}_1^m}, \tilde{s}_{\mathbb{N}_1^{m+1} \backslash \mathbb{N}_1^m}, a_1)) \max_{\tilde{a}_1} Q_t^{(m)}(s'_{\mathbb{N}_1^m}, \tilde{a}_1),
\end{aligned}
$$

$$(18)$$

where step (i) follows from the $R_1(s_1, a_1)$ term cancelling out as $m > 0$; step (ii) follows from the fact that each of the iterates $\{Q_t^{(m)}\}_{t \geq 0}$ only depends on the $m$-hop neighborhood of agent 1 as the projection operator fixes the states outside agent 1's $m$-hop neighborhood; and step (iii) follows from the decomposition (1c) of the transition kernel.

We now use the following lemma to upper bound the right-hand side and decouple the maximum. For each $\theta$ in some set $\Theta$, let $f_\theta : \mathcal{Y} \to \mathbb{R}$ denote a real-valued function on the Cartesian product space $\mathcal{Y} := \otimes_{i=1}^k \mathcal{Y}_i$.

**Lemma 3.** *For any pair $\mathbb{Q}^{(1)}$ and $\mathbb{Q}^{(2)}$ of product measures over $\mathcal{Y} = \otimes_{i=1}^k \mathcal{Y}_i$, we have*

$$
\sum_{y_1, \ldots, y_k} (\mathbb{Q}^{(1)}(y_1, \ldots, y_k) - \mathbb{Q}^{(2)}(y_1, \ldots, y_k)) \max_\theta f_\theta(y_1, \ldots, y_k) \leq \sum_i d_{TV}(\mathbb{Q}_i^{(1)}, \mathbb{Q}_i^{(2)}) \max_{y'_i, \tilde{y}_i, y_{-i}, \theta} |f_\theta(y'_i, y_{-i}) - f_\theta(\tilde{y}_i, y_{-i})|.
$$

Variants of this lemma without the maximum operation underlie the classical Dobrushin analysis of mixing times (Dobrushin, 1970), and have been exploited in previous work on graph-structured MDPs (Qu et al., 2020). See Section A.3 for the proof.

Using the representation (18) of the Bellman residual in conjunction with Lemma 3, we have

$$
\begin{aligned}
(\mathscr{B}^* Q_t^{(m)})(\boldsymbol{s}, a_1) - Q_{t+1}^{(m)}(\boldsymbol{s}, a_1) &\leq \sum_{\ell \in \mathbb{N}_1^m} d_{\mathrm{TV}}(\mathbb{P}_\ell(\cdot \mid s_{\mathbb{N}_1^{m+1}}, a_1), \mathbb{P}_\ell(\cdot \mid s_{\mathbb{N}_1^m}, \tilde{s}_{\mathbb{N}_1^{m+1} \backslash \mathbb{N}_1^m}, a_1)) \delta_\ell(Q_t^{(m)}) \\
&\leq \sum_{\ell \in \mathbb{N}_1^m} \mathbf{H}_{1,\ell}^{(m)} \delta_\ell(Q_t^{(m)}),
\end{aligned}
$$

$$(19)$$

where the final inequality follows from the definition of the $m$-hop influence matrix $\mathbf{H}^{(m)}$.

## A.2. Proof of Lemma 2

In order to prove this claim, we proceed via induction on the iteration number $t$. Beginning with $t = 0$, given our initialization $Q_0^{(m)} = \mathbf{0}$, we have $\delta_\ell(Q_0^{(m)}) = 0$, so that the bound (17) holds.

In the inductive step, we show that if the bound (17) holds for some $t$, then it also holds at time step $t + 1$. Using the update equation for $Q_{t+1}^{(m)}$, we can write

$$
Q_{t+1}^{(m)}(s_\ell, s_{\mathbb{N}_1^m \backslash \{\ell\}}, a_1) - Q_{t+1}^{(m)}(s'_\ell, s_{\mathbb{N}_1^m \backslash \{\ell\}}, a_1) = T_1 + T_2
$$

where $T_1 := \mathbb{I}(\ell = 1)(R_1(s_\ell, a_1) - R_1(s'_\ell, a_1))$, and

$$
T_2 := \gamma \sum_{\tilde{s}_{\mathbb{N}_1^m}} (\mathbb{P}(\tilde{s}_{\mathbb{N}_1^m} | s_\ell, s_{\mathbb{N}_1^{m+1} \backslash \{\ell\}}, a_1) - \mathbb{P}(\tilde{s}_{\mathbb{N}_1^m} | s'_\ell, s_{\mathbb{N}_1^{m+1} \backslash \{\ell\}}, a_1)) \max_{\tilde{a}_1} Q_t^{(m)}(\tilde{s}_{\mathbb{N}_1^m}, \tilde{a}_1).
$$

By definition of the oscillation, we have $|T_1| \leq \mathbb{I}(\ell = 1) \mathrm{osc}(R_1)$.

Applying Lemma 3 to $T_2$ yields

$$|T_2| \leq \gamma \max_{s,a_1,s'} \sum_{\ell} \sum_{j \in \mathbb{N}_1^m} d_{\mathrm{TV}}(\mathbb{P}_j(\cdot|s_\ell, s_{\mathbb{N}_1^{m+1} \setminus \{\ell\}}, a_1), \mathbb{P}_j(\cdot|s'_\ell, s_{\mathbb{N}_1^{m+1} \setminus \{\ell\}}, a_1)) \, \delta_j(Q^t) \leq \gamma \sum_{j \in \mathbb{N}_1^m} \mathbf{C}_{j,\ell} \delta_j(Q_t^{(m)}),$$

We have upper bounded the $Q^{t+1}$ difference for any choice of state-action, so we can take the maximum so as to bound $\delta_\ell(Q^{t+1})$. Doing so and combining our upper bounds on $|T_1|$ and $|T_2|$, we find that

$$\delta_\ell(Q_{t+1}^{(m)}) \leq \mathrm{osc}(R_1)\mathbb{I}(\ell = 1) + \gamma \sum_{j \in \mathbb{N}_1^m} \mathbf{C}_{j,\ell} \delta_j(Q_t^{(m)}). \tag{20}$$

The differences $\delta_j(Q_t^{(m)})$ can be upper bounded using the induction hypothesis. Substituting these upper bounds into inequality (20), we find that

$$\delta_\ell(Q_{t+1}^{(m)}) \leq \mathrm{osc}(R_1)\mathbb{I}(\ell = 1) + \gamma \sum_{j \in \mathbb{N}_1^m} \mathbf{C}_{j,\ell} \mathrm{osc}(R_1)\Big(\mathbb{I}(j = 1) + \sum_{r=1}^t \gamma^r (\mathbf{C}^r)_{(1,j)}\Big)$$

$$\leq \mathrm{osc}(R_1)\mathbb{I}(\ell = 1) + \gamma \mathrm{osc}(R_1)\mathbf{C}_{1,\ell} + \gamma \mathrm{osc}(R_1) \sum_{r=1}^{k-1} \gamma^r \sum_{j \in \mathbb{N}_1^m} \mathbf{C}_{j,\ell}(\mathbf{C}^r)_{(1,j)}$$

$$= \mathrm{osc}(R_1)\mathbb{I}(\ell = 1) + \gamma \mathrm{osc}(R_1)\mathbf{C}_{1,\ell} + \mathrm{osc}(R_1) \sum_{r=2}^{t+1} \gamma^r (\mathbf{C}^r)_{1,\ell}$$

$$= \mathrm{osc}(R_1)\Big\{\mathbb{I}(\ell = 1) + \sum_{r=1}^{t+1} \gamma^r (\mathbf{C}^r)_{1,\ell}\Big\},$$

which completes the proof.

### A.3. Proof of Lemma 3

Define the sequence of product measures

$$\mathbb{Q}^{(i)} := \Big(\bigotimes_{j=1}^i \mathbb{Q}_j^{(1)}\Big) \otimes \Big(\bigotimes_{j=i+1}^k \mathbb{Q}_j^{(2)}\Big), \qquad \text{for } i = 0, 1, \ldots, k.$$

As the index $i$ increases from 0 to $k$, the measure $\mathbb{Q}^{(i)}$ is transformed from $\bigotimes_{j=1}^k \mathbb{Q}_j^{(2)}$ into $\bigotimes_{j=1}^k \mathbb{Q}_j^{(1)}$ on a product-by-product basis. Using this sequence, we have the telescoping relation

$$\bigotimes_{j=1}^k \mathbb{Q}_j^{(1)} - \bigotimes_{j=1}^k \mathbb{Q}_j^{(2)} = \sum_{i=1}^k (\mathbb{Q}^{(i)} - \mathbb{Q}^{(i-1)}),$$

from which it follows that

$$\sum_{y_1,\ldots,y_k} (\mathbb{Q}^{(1)} - \mathbb{Q}^{(2)})(y_1,\ldots,y_k) \max_\theta f_\theta(y_1,\ldots,y_k) = \sum_{i=1}^k \sum_{y_1,\ldots,y_k} (\mathbb{Q}^{(i)} - \mathbb{Q}^{(i-1)})(y_1,\ldots,y_k) \max_\theta f_\theta(y_1,\ldots,y_k).$$

For any given coordinate $i$, since $\mathbb{Q}^{(i)}$ and $\mathbb{Q}^{(i-1)}$ differ only in the $i$-th coordinate, we can write

$$(\mathbb{Q}^{(i)} - \mathbb{Q}^{(i-1)})(y) = (\mathbb{Q}_i^{(1)} - \mathbb{Q}_i^{(2)})(y_i) \prod_{j<i} \mathbb{Q}_j^{(1)}(y_j) \prod_{j>i} \mathbb{Q}_j^{(2)}(y_j).$$

Using this representation, we can write

$$\sum_{y_1,\ldots,y_k} (\mathbb{Q}^{(i)} - \mathbb{Q}^{(i-1)})(y) \max_\theta f_\theta(y) = \sum_{y_{-i}} \prod_{j<i} \mathbb{Q}_j^{(1)}(y_j) \prod_{j>i} \mathbb{Q}_j^{(2)}(y_j) \sum_{y_i} (\mathbb{Q}_i^{(1)} - \mathbb{Q}_i^{(2)})(y_i) \max_\theta f_\theta(y_i, y_{-i}),$$

Since $\sum_{y_i} (\mathbb{Q}_i^{(1)} - \mathbb{Q}_i^{(2)})(y_i) = 0$, we have

$$\sum_{y_i} (\mathbb{Q}_i^{(1)} - \mathbb{Q}_i^{(2)})(y_i) \max_\theta f_\theta(y_i, y_{-i}) = \sum_{y_i} (\mathbb{Q}_i^{(1)} - \mathbb{Q}_i^{(2)})(y_i) \left( \max_\theta f_\theta(y_i, y_{-i}) - \min_{\tilde{y}_i} \max_\theta f_\theta(\tilde{y}_i, y_{-i}) \right)$$

$$\overset{(i)}{\leq} \sum_{y_i : \mathbb{Q}_i^{(1)} \geq \mathbb{Q}_i^{(2)}} (\mathbb{Q}_i^{(1)} - \mathbb{Q}_i^{(2)})(y_i) \left( \max_\theta f_\theta(y_i, y_{-i}) - \min_{\tilde{y}_i} \max_\theta f_\theta(\tilde{y}_i, y_{-i}) \right)$$

$$\overset{(ii)}{\leq} d_{\mathrm{TV}}(\mathbb{Q}_i^{(1)}, \mathbb{Q}_i^{(2)}) \max_{y_i, \tilde{y}_i, y_{-i}} \left| \max_\theta f_\theta(y_i, y_{-i}) - \max_\theta f_\theta(\tilde{y}_i, y_{-i}) \right|,$$

where step (i) follows from dropping non-positive terms; and step (ii) follows from the definition $d_{\mathrm{TV}}(\mathbb{P}, \mathbb{Q}) = \sum_{u | \mathbb{P}(u) \geq \mathbb{Q}(u)} (\mathbb{P}(u) - \mathbb{Q}(u))$ of the TV distance between measures $\mathbb{P}$ and $\mathbb{Q}$.

Taking the expectation over $y_{-i}$ and upper bounding by the maximum yields

$$\sum_{y_1, \ldots, y_k} (\mathbb{Q}^{(i)} - \mathbb{Q}^{(i-1)})(y) \max_\theta f_\theta(y) \leq d_{\mathrm{TV}}(\mathbb{Q}_i^{(1)}, \mathbb{Q}_i^{(2)}) \max_{y_i', \tilde{y}_i, y_{-i}, \theta} \left| f_\theta(y_i', y_{-i}) - f_\theta(\tilde{y}_i, y_{-i}) \right|.$$

Summing over $i = 1, \ldots, k$ completes the proof.

## B. Proof of Theorem 2

Here we prove our guarantee for the multi-agent setting. Our proof uses the following standard result (Bertsekas, 2022): given any function $Q$ and a policy $\boldsymbol{\pi}$ that is greedy with respect to $Q$, the value function $Q^{\boldsymbol{\pi}}$ of the policy $\boldsymbol{\pi}$ satisfies the bound

$$\|Q^{\boldsymbol{\pi}} - Q^*\|_\infty \leq \frac{2\gamma}{1 - \gamma} \|Q - Q^*\|_\infty. \tag{21}$$

This bound reduces our problem to defining a $Q$-function that approximates $Q^*$ well, and our argument consists of three main steps.

**Telescoping decomposition:**  Our first step is to decompose $Q^*$ as a telescoping sum, one which exposes the effects of individual coordinates. For each node $i \in [n]$, we define the *coordinate effect function*

$$\delta_i(Q^*)(\boldsymbol{s}, \boldsymbol{a}; \overline{s}_i, \overline{a}_i) := Q^*((s, a)_i, (s, a)_{-i}) - Q^*((\overline{s}, \overline{a})_i, (s, a)_{-i}), \tag{22a}$$

which measures how changes in the state-action pair at node $i$ affect the optimal $Q$-function. In terms of this notation, for any fixed global state-action pair $(\overline{s}, \overline{a})$, we have the telescoping decomposition

$$Q^*(\boldsymbol{s}, \boldsymbol{a}) = \sum_{i=1}^n \delta_i(Q^*)((\overline{s}, \overline{a})_{<i}, s_i, a_i, (s, a)_{>i} ; \overline{s}_i, \overline{a}_i) + Q^*(\overline{s}, \overline{a}). \tag{22b}$$

For each $i \in [n]$, let $\Pi_i^{(m/2)}$ be the projection operator that fixes all states and actions outside $\mathbb{N}_i^{m/2}$ to $(\overline{s}, \overline{a})$. With this notatation, our $m$-hop cooperative approximation to the optimal $Q$-function $Q^*$ is given by

$$\widetilde{Q}^{(m)}(\boldsymbol{s}, \boldsymbol{a}) := \sum_{i=1}^n \Pi_i^{(m/2)} \delta_i(Q^*)((\overline{s}, \overline{a})_{<i}, s_i, a_i, (s, a)_{>i} ; \overline{s}_i, \overline{a}_i) + Q^*(\overline{s}, \overline{a}). \tag{22c}$$

By comparison of equations (22b) and (22c), we have $(\widetilde{Q}^{(m)} - Q^*)(\boldsymbol{s}, \boldsymbol{a}) = \sum_{i=1}^n F_i$, where

$$F_i := \Pi_i^{(m/2)} \delta_i(Q^*)((\overline{s}, \overline{a})_{<i}, s_i, a_i, (s, a)_{>i} ; \overline{s}_i, \overline{a}_i) - \delta_i(Q^*)((\overline{s}, \overline{a})_{<i}, s_i, a_i, (s, a)_{>i} ; \overline{s}_i, \overline{a}_i). \tag{22d}$$

**From coordinate effect to correlation gap:** Note that the term $\|(\widetilde{Q}^{(m)} - Q^*)\|_\infty = \|\sum_{i=1}^n F_i\|_\infty$ corresponds to a sum of approximation errors, one for each coordinate effect. Our next step is to upper bound this sum in terms of a pairwise measure. In particular, for any pair of distinct agents $i$ and $j$, we define the *decorrelation gap*

$$\Delta_{i,j}(Q^*) := \max_{\substack{s,a, \\ \tilde{s},\tilde{a}}} |\delta_i(Q^*)(\boldsymbol{s}, \boldsymbol{a}; \tilde{s}_i, \tilde{a}_i) - \delta_i(Q^*)((\tilde{s}, \tilde{a})_j, (s, a)_{-j}; \tilde{s}_i, \tilde{a}_i)|, \tag{23a}$$

i.e., the maximum effect of changing the local state-action of agent $j$ on the coordinate effect $\delta_i$ of agent $i$. The following lemma relates the error $\|\widetilde{Q}^{(m)} - Q^*\|_\infty$ to these decorrelation gaps, and the pairwise influence matrices.

**Lemma 4** (From $Q$-error to decorrelation gaps). *We have the bound*

$$\|\widetilde{Q}^{(m)} - Q^*\|_\infty \leq \gamma \sum_{i \in [n]} \sum_{j \in \left(\mathbb{N}_i^{m/2-1}\right)^c} \sum_{\ell \in \mathbb{N}_i^1} \mathbf{H}_{i,j}^{(m)} \mathbf{C}_{i,\ell} \Delta_{\ell,j}(Q^*). \tag{23b}$$

See Section B.1 for the proof.

**Completing the proof:** In the third part of the proof, we bound the decorrelation gaps, and then combine with inequality (23b) to complete the argument. Recall the definition (10a) of the two-hop adjacency matrix $\mathbf{W}$ along with the unique solution $\mathbf{D}$ to the discrete-time Lyapunov equation (10b). With this notation, we have

**Lemma 5.** *Under the spectral decay condition $\sqrt{\gamma}\rho(\mathbf{C}) < 1$, the decorrelation gap between any pair of agents $i$ and $j$ can be bounded as*

$$\Delta_{i,j}(Q^*) \leq \frac{2\mathrm{osc}(R)}{1 - \gamma} \left( (\sqrt{\gamma}\mathbf{C}^\intercal)^{(\lfloor d_{\mathcal{G}}(i,j)/2 \rfloor - 1)_+} \mathbf{D}(\sqrt{\gamma}\mathbf{C})^{(\lfloor d_{\mathcal{G}}(i,j)/2 \rfloor - 1)_+} \right)_{(i,j)}. \tag{24}$$

Lemma 5 shows that as the distance between nodes $i$ and $j$ increases, the decorrelation gaps have a decay property as a function of $\sqrt{\gamma}\mathbf{C}$.

The final step in the proof is to substitute the bounds on the decorrelation gaps from Lemma 5 in the right-hand side of the bound in Lemma 4, giving

$$\|\widetilde{Q}^{(m)} - Q^*\|_\infty \leq \frac{2\gamma\mathrm{osc}(R)}{1 - \gamma} \sum_{i \in [n]} \sum_{j \in \left(\mathbb{N}_i^{m/2-1}\right)^c} \mathbf{H}_{i,j}^{(m)} (\mathbf{C}(\sqrt{\gamma}\mathbf{C}^\intercal)^{(\lfloor m/4 \rfloor - 1)_+} \mathbf{D}(\sqrt{\gamma}\mathbf{C})^{(\lfloor m/4 \rfloor - 1)_+})_{(i,j)}.$$

Plugging in this approximation error bound into the bound (21) completes the proof of existence of an $m$-hop cooperative policy.

It remains to prove our two auxiliary results (Lemmas 4 and 5). In order to do so, we make use of the following result, which applies to probability measures $\mathbb{P}_X, \mathbb{Q}_X$ over the space $\mathcal{X}$, measures $\mathbb{P}_Y, \mathbb{Q}_Y$ over the space $\mathcal{Y}$, and $\mathbb{P}_Z$ over the space $\mathcal{Z}$.

**Lemma 6** (Decoupling Maximum). *Given a parameterized family $\{f_\theta \mid \theta \in \Theta\}$ of functions $(x, y, z) \mapsto f_\theta(x, y, z) \in \mathbb{R}$, we have*

$$\sum_{x,y,z} \mathbb{P}_Z(z)(\mathbb{P}_X(x) - \mathbb{Q}_X(x))(\mathbb{P}_Y(y) - \mathbb{Q}_Y(y)) \max_\theta f_\theta(x, y, z) \leq d_{TV}(\mathbb{P}_X, \mathbb{Q}_X)\, d_{TV}(\mathbb{P}_Y, \mathbb{Q}_Y)\, \tilde{\Delta}(f), \tag{25}$$

*where $\tilde{\Delta}(f) := \max_{\substack{x,y,z, \\ \overline{x},\overline{y},\theta}} \left| \{f_\theta(x, y, z) - f_\theta(x, \overline{y}, z)\} - \{f_\theta(\overline{x}, y, z) - f_\theta(\overline{x}, \overline{y}, z)\} \right|.$*

See Section B.3 for the proof.

## B.1. Proof of Lemma 4

By the definition, the optimal $Q$-function satisfies the Bellman equation

$$Q^*(\boldsymbol{s}, \boldsymbol{a}) = R(\boldsymbol{s}, \boldsymbol{a}) + \gamma \sum_{\tilde{\boldsymbol{s}} \in \mathcal{S}} \mathbb{P}(\tilde{\boldsymbol{s}} \mid \boldsymbol{s}, \boldsymbol{a}) \max_{\tilde{\boldsymbol{a}} \in \mathcal{A}} Q^*(\tilde{\boldsymbol{s}}, \tilde{\boldsymbol{a}}).$$

Therefore, the one-step differences $\delta_i(Q^*) \equiv Q^*((\overline{s}, \overline{a})_{<i}, s_i, a_i, (s, a)_{>i}) - Q^*(\overline{s}_{\leq i}, \overline{a}_{\leq i}, (s, a)_{>i})$ can be written as

$$\delta_i(Q^*) = R_i(s_i, a_i) - R_i(\overline{s}_i, \overline{a}_i) + \gamma \sum_{\tilde{s}} \left( \mathbb{P}(\tilde{s} \mid (\overline{s}, \overline{a})_{<i}, s_i, a_i, (s, a)_{>i}) - \mathbb{P}(\tilde{s} \mid \overline{s}_{\leq i}, \overline{a}_{\leq i}, (s, a)_{>i}) \right) \max_{\tilde{a} \in \mathcal{A}} Q^*(\tilde{s}, \tilde{a}).$$

Subtracting $\delta_i(Q^*)$ from its projected version (22d) gives

$$F_i = \gamma \sum_{\tilde{s}} \left( \Pi_i^{(m/2)} \left( \mathbb{P}(\tilde{s} \mid (\overline{s}, \overline{a})_{<i}, s_i, a_i, (s, a)_{>i}) - \mathbb{P}(\tilde{s} \mid \overline{s}_{\leq i}, \overline{a}_{\leq i}, (s, a)_{>i}) \right) \right.$$
$$\left. - \left( \mathbb{P}(\tilde{s} \mid (\overline{s}, \overline{a})_{<i}, s_i, a_i, (s, a)_{>i}) - \mathbb{P}(\tilde{s} \mid \overline{s}_{\leq i}, \overline{a}_{\leq i}, (s, a)_{>i}) \right) \right) \max_{\tilde{a} \in \mathcal{A}} Q^*(\tilde{s}, \tilde{a}).$$

Since the above difference in the transition function changes only in the state and action of agent $i$, only the transition kernels for nodes in the neighborhood of state $i$ will be affected due to the decomposition in the transition kernel (1c). Additionally, as we assume that $m/2 > 1$, fixing the states outside the $m/2$ neighborhood, as we do in $F_i$, will not affect the transition kernels of the nodes in the neighborhood of state $i$. Therefore we have

$$F_i = \gamma \sum_{\tilde{s}} \mathbb{P}(s_{\mathbb{N}_i^{m/2-1} \cap (\mathbb{N}_i^1)^c} \mid (\overline{s}, \overline{a})_{<i}, (\overline{s}, \overline{a})_{>i}) \left( \mathbb{P}(s_{\mathbb{N}_i^1} \mid (\overline{s}, \overline{a})_{<i}, s_i, a_i, (s, a)_{>i}) - \mathbb{P}(s_{\mathbb{N}_i^1} \mid \overline{s}_{\leq i}, \overline{a}_{\leq i}, (s, a)_{>i}) \right)$$
$$\left( \Pi_i^{(m/2)} \mathbb{P}(s_{(\mathbb{N}_i^{m/2-1})^c} \mid (\overline{s}, \overline{a})_{<i}, s_i, a_i, (s, a)_{>i}) - \mathbb{P}(s_{(\mathbb{N}_i^{m/2-1})^c} \mid \overline{s}_{\leq i}, \overline{a}_{\leq i}, (s, a)_{>i}) \right) \max_{\tilde{a} \in \mathcal{A}} Q^*(\tilde{s}, \tilde{a})$$

Using Lemma 6 then gives the bound

$$\|F_i\|_\infty \leq \gamma \sum_{j \in (\mathbb{N}_i^{m/2-1})^c} \sum_{\ell \in \mathbb{N}_i^1} \mathbf{H}_{i,j}^{(m)} \mathbf{C}_{i,\ell} \Delta_{\ell,j}.$$

## B.2. Proof of Lemma 5

We consider the Q-iteration $Q^t = \mathscr{B}^* Q^{t-1}$ which we know converges to the optimal Q-function $Q^*$. Define coordinate effects $\delta_i^t$ as

$$\delta_i^t(s, a; \overline{s}_i, \overline{a}_i) := Q^t((s, a)_i, (s, a)_{-i}) - Q^t((\overline{s}, \overline{a})_i, (s, a)_{-i})$$

and the sequence of matrices $\Delta^t$ similarly. Assume that $\mathbb{N}_i^1 \cap \mathbb{N}_j^1 = \emptyset$, i.e., agents $i$ and $j$ are at least three hops away from each other on the graph. Then

$$\delta_i^t(s, a; \overline{s}_i, \overline{a}_i) - \delta_i^t((\overline{s}, \overline{a})_j, (s, a)_{-j}; \overline{s}_i, \overline{a}_i) = \gamma \sum_{s'} \left( \mathbb{P}(s' \mid s, a) - \mathbb{P}(s' \mid (\overline{s}, \overline{a})_i, (s, a)_{-i}) \right) \max_{a'} Q^{t-1}(s', a')$$
$$- \gamma \sum_{s'} \left( \mathbb{P}(s' \mid (\overline{s}, \overline{a})_j, (s, a)_{-j}) - \mathbb{P}(s' \mid (\overline{s}, \overline{a})_{i,j}, (s, a)_{-\{i,j\}}) \right) \max_{a'} Q^{t-1}(s', a'). \tag{26}$$

The reward terms cancel since $i \neq j$. Recall that from the decomposition we have for the transition kernel (1c), the effect of changing the state-action of agent $i$ in the above expression will arise only in the transition kernel of its neighborhood, i.e.,

$$\left( \mathbb{P}(s' \mid s, a) - \mathbb{P}(s' \mid (\overline{s}, \overline{a})_i, (s, a)_{-i}) \right) \max_{a'} Q^{t-1}(s', a')$$
$$= \sum_{s'} \prod_{q \in \mathbb{N}_i^{1c}} \mathbb{P}_q(s_q' \mid (s, a)_{\mathbb{N}_q^1}) \left( \prod_{q \in \mathbb{N}_i^1} \mathbb{P}_q(s_q' \mid (s, a)_{\mathbb{N}_q^1}) - \prod_{q \in \mathbb{N}_i^1} \mathbb{P}_q(s_q' \mid (\overline{s}, \overline{a})_i, (s, a)_{\mathbb{N}_q^1 \setminus \{-i\}}) \right) \max_{a'} Q^{t-1}(s', a'). \tag{27}$$

Let $I_i$ be an increasing ordering for the set $\mathbb{N}_i^1$. In the difference of product terms above, we first add and subtract terms that only differ in one product term in the following way:

$$
\left( \prod_{q \in \mathbb{N}_i^1} \mathbb{P}_q\big(s_q' \mid (s,a)_{\mathbb{N}_q^1}\big) - \prod_{q \in \mathbb{N}_i^1} \mathbb{P}_q\big(s_q' \mid (\overline{s}_i,\overline{a}_i),(s,a)_{\mathbb{N}_q^1 \setminus \{i\}}\big) \right) \max_{\boldsymbol{a}'} Q^{t-1}(\boldsymbol{s}',\boldsymbol{a}')
$$

$$
= \sum_{u \in I_i} \left[ \prod_{\substack{v \in I_i \\ v > u}} \mathbb{P}_v\big(s_v' \mid (s,a)_{\mathbb{N}_v^1}\big) \prod_{\substack{v \in I_i \\ v < u}} \mathbb{P}_v\big(s_v' \mid (\overline{s}_i,\overline{a}_i),(s,a)_{\mathbb{N}_v^1 \setminus \{i\}}\big) \right.
$$

$$
\left. \Big( \mathbb{P}_u\big(s_u' \mid (s,a)_{\mathbb{N}_u^1}\big) - \mathbb{P}_u\big(s_u' \mid (\overline{s}_i,\overline{a}_i),(s,a)_{\mathbb{N}_u^1 \setminus \{i\}}\big) \Big) \right] \max_{\boldsymbol{a}'} Q^{t-1}(\boldsymbol{s}',\boldsymbol{a}'). \tag{28}
$$

Now that we have characterized the effect of changing agent $i$'s state-action pair, we incorporate the effect of changing agent $j$'s state-action pair. Since $\mathbb{N}_i^1 \cap \mathbb{N}_j^1 = \emptyset$, the effect of changing agent $j$ lies entirely in the transition kernels of agents in $\mathbb{N}_j^1 \subset \mathbb{N}_i^{1\,\mathsf{c}}$. Employing the same add-and-subtract argument, we obtain

$$
\delta_i^t(\boldsymbol{s},\boldsymbol{a};\overline{s}_i,\overline{a}_i) - \delta_i^t((\overline{s},\overline{a})_j,(s,a)_{-j};\overline{s}_i,\overline{a}_i) = \gamma \sum_{u \in I_i} \sum_{w \in I_j} \sum_{\boldsymbol{s}'} \prod_{q \in \mathbb{N}_i^{1\,\mathsf{c}} \cap \mathbb{N}_j^{1\,\mathsf{c}}} \mathbb{P}_q\big(s_q' \mid (s,a)_{\mathbb{N}_q^1}\big)
$$

$$
\times \prod_{\substack{v \in I_i \\ v > u}} \mathbb{P}_v\big(s_v' \mid (s,a)_{\mathbb{N}_v^1}\big) \prod_{\substack{v \in I_i \\ v < u}} \mathbb{P}_v\big(s_v' \mid (\overline{s}_i,\overline{a}_i),(s,a)_{\mathbb{N}_v^1 \setminus \{i\}}\big)
$$

$$
\times \prod_{\substack{v \in I_j \\ v > w}} \mathbb{P}_v\big(s_v' \mid (s,a)_{\mathbb{N}_v^1}\big) \prod_{\substack{v \in I_j \\ v < w}} \mathbb{P}_v\big(s_v' \mid (\overline{s}_j,\overline{a}_j),(s,a)_{\mathbb{N}_v^1 \setminus \{j\}}\big)
$$

$$
\times \Big( \mathbb{P}_u\big(s_u' \mid (s,a)_{\mathbb{N}_u^1}\big) - \mathbb{P}_u\big(s_u' \mid (\overline{s}_i,\overline{a}_i),(s,a)_{\mathbb{N}_u^1 \setminus \{i\}}\big) \Big)
$$

$$
\times \Big( \mathbb{P}_w\big(s_w' \mid (s,a)_{\mathbb{N}_w^1}\big) - \mathbb{P}_w\big(s_w' \mid (\overline{s}_j,\overline{a}_j),(s,a)_{\mathbb{N}_w^1 \setminus \{j\}}\big) \Big) \max_{\boldsymbol{a}'} Q^{t-1}(\boldsymbol{s}',\boldsymbol{a}'). \tag{29}
$$

Our next step is to apply Lemma 6 to each node in the one-hop neighborhoods of agents $i$ and $j$. The product of differences in Lemma 6 is created by the product of differences for each pair of nodes in the one-hop neighborhoods of $i$ and $j$, with $f_\theta$ being the iterate $Q^{t-1}$ parameterized by the actions. Doing so yields

$$
\delta_i^t(\boldsymbol{s},\boldsymbol{a};\overline{s}_i,\overline{a}_i) - \delta_i^t((\overline{s},\overline{a})_j,(s,a)_{-j};\overline{s}_i,\overline{a}_i) \leq \gamma \sum_{u \in I_i} \sum_{w \in I_j} \mathbf{C}_{u,i}\mathbf{C}_{w,j}\Delta_{u,w}^{t-1}, \tag{30a}
$$

Combining this inequality with equation (30a) leads to the recursive bound

$$
\Delta_{i,j}^t \leq \gamma \sum_{u \in I_i} \sum_{w \in I_j} \mathbf{C}_{u,i}\mathbf{C}_{w,j}\Delta_{u,w}^{t-1}. \tag{30b}
$$

We can then upper bound $\Delta^k$ by solving this recursive inequality.

Rewriting the bound (30b) in terms of matrices, we obtain

$$
\Delta^t \preceq_{\text{e.w.}} \frac{2\text{osc}(R)}{1-\gamma}\mathbf{W} + \gamma\mathbf{C}^T\Delta^{t-1}\mathbf{C}, \tag{30c}
$$

where $\preceq_{\text{e.w.}}$ is the element-wise partial order. The entries of the matrix $\mathbf{W}$ are a binary indicator for whether the corresponding nodes are within two hops of each other. When they are within two hops of each other, we bound the corresponding entry of $\Delta^t$ with the trivial bound $2\text{osc}(R)/(1-\gamma)$.

Since the elements of $\mathbf{C}$ are nonnegative, the bound (30c) can be recursed in terms of the $\preceq_{\text{e.w.}}$ partial order. It follows that

$$
\Delta := \limsup_{t \uparrow \infty} \Delta^t \preceq_{\text{e.w.}} \limsup_{t \uparrow \infty} \frac{2\text{osc}(R)}{1-\gamma} \sum_{r=0}^{t-1} \gamma^r (\mathbf{C}^\intercal)^r \mathbf{W} \mathbf{C}^r.
$$

Note that for agents $i$ and $j$, the first $\lfloor d_{\mathcal{G}}(i,j)/2 \rfloor - 1$ terms in the above sum will be zero. This is because the matrix $\mathbf{C}$ is a weighted adjacency matrix, resulting in the entries of $\mathbf{C}^r$ being the sum of the weights incurred in any $r$-hop path between the nodes corresponding to that entry. Then with the assumption $\sqrt{\gamma}\rho(\mathbf{C}) < 1$, it follows that (Horn & Johnson, 2012)

$$\Delta_{i,j} \leq \frac{2\delta(R)}{1-\gamma}(\sqrt{\gamma}\mathbf{C}^{\mathsf{T}})^{(\lfloor d_{\mathcal{G}}(i,j)/2 \rfloor - 1)_+}\mathbf{D}(\sqrt{\gamma}\mathbf{C})^{(\lfloor d_{\mathcal{G}}(i,j)/2 \rfloor - 1)_+},$$

### B.3. Proof of Lemma 6

The main crux of the proof is noting that in the left-hand side of the bound (25), adding any term that depends only on $(x,z)$ or only on $(y,z)$ to the maximum over $f_\theta$ has no effect due to the difference in expectations in both $x$ and $y$. Recall that the TV distance between distributions $\mathbb{P}$ and $\mathbb{Q}$ can be written as $d_{\mathrm{TV}}(\mathbb{P},\mathbb{Q}) = \sum_{u|\mathbb{P}(u)\geq\mathbb{Q}(u)}\big(\mathbb{P}(u) - \mathbb{Q}(u)\big)$. Consequently, for any function $g: \mathcal{U} \to \mathbb{R}$, we have the bound

$$\mathbb{E}_{\mathbb{P}}[g] - \mathbb{E}_{\mathbb{Q}}[g] = \mathbb{E}_{\mathbb{P}}[g - g_{\min}] - \mathbb{E}_{\mathbb{Q}}[g - g_{\min}] \;\leq\; d_{\mathrm{TV}}(\mathbb{P},\mathbb{Q})\max_{u,\overline{u}}\big(g(u) - g(\overline{u})\big). \tag{31a}$$

where $g_{\min} := \min_{u\in\mathcal{U}} g(u)$. By symmetry, the same bound also holds for the difference $\mathbb{E}_{\mathbb{Q}}[g] - \mathbb{E}_{\mathbb{P}}[g]$.

Using the bound (31a) only on the inner sum for the variable $y$, we obtain

$$\sum_{x,y,z} \mathbb{P}_Z(z)(\mathbb{P}_X(x) - \mathbb{Q}_X(x))(\mathbb{P}_Y(y) - \mathbb{Q}_Y(y))\max_\theta f_\theta(x,y,z)$$

$$\leq d_{\mathrm{TV}}(\mathbb{P}_Y,\mathbb{Q}_Y)\sum_{x,z}\mathbb{P}_Z(z)(\mathbb{P}_X(x) - \mathbb{Q}_X(x))\max_{y,\overline{y}}\big(\max_\theta f_\theta(x,y,z) - \max_\theta f_\theta(x,\overline{y},z)\big),$$

$$\leq d_{\mathrm{TV}}(\mathbb{P}_Y,\mathbb{Q}_Y)\sum_{x,z}\mathbb{P}_Z(z)(\mathbb{P}_X(x) - \mathbb{Q}_X(x))\max_{y,\overline{y},\theta}\big(f_\theta(x,y,z) - f_\theta(x,\overline{y},z)\big), \tag{31b}$$

where the last step follows from $\sup h - \sup g \leq \sup(h - g)$. Next, we repeat the same trick to obtain a difference term for the $x$ variable. Using the bound (31a) for the inner sum on $x$ gives

$$\sum_{x,z}\mathbb{P}_Z(z)(\mathbb{P}_X(x) - \mathbb{Q}_X(x))\max_{y,\overline{y},\theta}\big(f_\theta(x,y,z) - f_\theta(x,\overline{y},z)\big)$$

$$\leq d_{\mathrm{TV}}(\mathbb{P}_X,\mathbb{Q}_X)\max_{x,\overline{x}}\Big(\max_{y,\overline{y},\theta}\big(f_\theta(x,y,z) - f_\theta(x,\overline{y},z)\big) - \max_{y,\overline{y},\theta}\big(f_\theta(\overline{x},y,z) - f_\theta(\overline{x},\overline{y},z)\big)\Big)$$

$$\leq d_{\mathrm{TV}}(\mathbb{P}_X,\mathbb{Q}_X)\max_{x,\overline{x},y,\overline{y},\theta}\Big(\big(f_\theta(x,y,z) - f_\theta(x,\overline{y},z)\big) - \big(f_\theta(\overline{x},y,z) - f_\theta(\overline{x},\overline{y},z)\big)\Big). \tag{31c}$$

Combining the results of equations (31b) and (31c) yields the claim (25) of the lemma.

