# OpenReview forum: "Local Policies for Graph-Structured Markov Decision Processes"
_ICML.cc/2026/Conference — ICML 2026 regular_

### Official Review · Reviewer_XV6n · 2026-03-09

**Soundness:** 3
**Presentation:** 3
**Significance:** 3
**Originality:** 3
**Overall Recommendation:** 5
**Confidence:** 4

**Summary:**

The paper introduces a methodology to compute approximately optimal cooperative policies on sparse graphs with sufficiently weak interactions between nodes. In such a scenario, due to sparsity the authors are able to achieve a result where policies based on only local states are sufficient for approximate optimality, even with partial observability. For the multi-agent cooperation scenario, they show the existence of such approximately optimal local policies, while an efficient learning algorithm for finding such policies is left to future work. The result is quite general and can apply to a variety of scenarios.

**Compliance With Llm Reviewing Policy:**

Affirmed.

**Final Justification:**

My concerns have been fully addressed. I appreciate the additional work on the multi-agent setting, and I think even just a brief discussion of initial results or possibilities in the appendix would significantly strengthen the paper. I will raise my score to 5.

**Key Questions For Authors:**

I just have a few quick questions to verify my understanding.

1.  I can see that the spectral radius is used for theoretical results and sparsity is required. What happens in the case of increasingly large graphs, such as via large graph limits?

2.  What happens when the assumption is not fulfilled? Does the methodology still degrade gracefully, indicating possibility of relaxing assumptions further?

**Limitations:**

yes

**Strengths And Weaknesses:**

1.  Originality: The core idea of applying exponential correlation decay and spectral radii to control / RL problems on sufficiently sparse graphs is original, though the work seems quite related to many recent results on local interaction / policies on large sparse graphs [1-3].

2.  Soundness: The paper is backed by theory, including rigorous approximate optimality results for local policies. In particular, the results seem plausible to me and are of similar character as other recent results in sparse graph literature. This provides a solid foundation for any empirical results.

3.  Significance: The experiments and the algorithmic contribution are a little limited, given that there is no proposed algorithm for the multi-agent scenario. But the experiment does verify the theory quite nicely.

4.  Presentation: The paper positions itself primarily within a growing line of recent work on graph-structured MDPs and RL. I think the work also seems quite related to many recent results on local sparse interactions. Overall, the work is written well and very clear on its results.

[1] Lacker et al. "Local weak convergence for sparse networks of interacting processes." The Annals of Applied Probability 33.2 (2023): 843-888.

[2] Ankan Ganguly and Kavita Ramanan. "Hydrodynamic limits of non-Markovian interacting particle systems on sparse graphs." Electronic Journal of Probability 29 (2024): 1-63.

[3] Fabian et al. "Learning mean field control on sparse graphs." ICML 2025.

---

> ### Author Rebuttal · Authors · 2026-03-31
>
> We thank the reviewer for their questions and comments.
>
> __Relation to works [1-3]:__ Regarding the references [1-3] cited, they all study a mean field regime where the number of agents $n$ goes to infinity, while we study a fixed $n$ regime, which are fundamentally different problems. Moreover, the works [1] and [2] study dynamics with no control and prove spatial localization results (i.e., the influence of distant nodes vanishes asymptotically). In contrast, we consider a fixed finite $n$ setting and study controlled dynamics (MDPs), where the goal is to approximate the globally optimal policy for a certain reward function.
>
> The work [3] studies mean-field control for large-scale systems, where the control of each agent is restricted to depend only on the states of its neighborhood. In contrast, our results show that such purely local policies can be suboptimal, and that achieving near-optimal performance may require cooperative policies, in which agents coordinate their actions over their $m$-hop neighborhoods. Our main result (Theorem 2) establishes that the globally optimal policy can be approximated by such $m$-hop cooperative policies with explicit non-asymptotic error bounds relative to the optimal $Q^*$.
>
> __Multi-agent setting:__ We agree that algorithmic guarantees for this setting is an important facet of this problem. We would like to highlight that our current result here (Theorem 2) already goes beyond the scope of prior work, which either restrict to (possibly suboptimal) local policies that only use local state information without any coordination, or use large amounts of regularization to make local policies near-optimal. It shows how local coordination across $m$-hop neighborhoods can be utilized to achieve approximately optimal global behavior.
>
> Since the submission, we have made progress on the algorithmic side for the multi-agent setting. We consider a projected Bellman iteration using the projection operator defined in Appendix B.2. We show that this projected Bellman iteration converges exponentially fast to a ball around $Q^*$, with the radius of the ball decaying exponentially in $m$. This result requires a stronger version of the spectral radius bound, i.e., $\sqrt{\gamma} n \rho(C)<1$, where $n$ is the number of players. The dependence on $n$ arises due to the need for controlling the projection errors of each of the $n$ players. It's not clear to us if we have the space to add this result to the current submission, but if the reviewers think it necessary, we would be happy to oblige or provide more details.
>
> __Bounded spectral radius assumption:__ In section 5.3, we discuss an upper bound on the spectral radius $\rho(C) \leq d_{max} C_{max}$. This highlights that the key factor in controlling the spectral radius is not exactly the size of the MDP, but rather the connectivity and interaction strength of the underlying graph. In particular, the condition can hold even for large systems, provided the graph is sufficiently sparse or the interactions are not too strong. This bound also reveals a natural trade-off between graph degree and interaction strength: higher connectivity requires weaker pairwise influences for the condition to hold. Therefore, sparsity is not a strong requirement.
>
> In sparse graphs, for this condition to not hold, the transitions of each local state would have to become almost deterministic. Here is such an example. Consider the case where agents 1 to $n$ are arranged on a line, such that only agent 1 has a non-zero reward function, but all agents have actions. It is possible to construct cases where, if all the transitions were to be deterministic, agent 1 needs to coordinate its action with agent $n$ to be optimal, and if it didn't, agent 1's local state goes into a permanent bad state such that the system incurs a very suboptimal reward.
>
> In the case of large graph limits, if $d_{max}$ and the max interaction strength $C_{max}$ do not grow with $n$, then the results would still hold. Our proof techniques ultimately require control on matrix sums of the form $\sum_{k \geq 0} \gamma^k C^k$. It could be that under large graph limits, this sum could be controlled for a majority of nodes (i.e., for a submatrix of $C$) under a looser condition that takes into account more information about the local properties of the nodes and the graph. This would require more careful analysis.
>
>
> [1] Lacker et al. "Local weak convergence for sparse networks of interacting processes." The Annals of Applied Probability 33.2 (2023): 843-888.
>
> [2] Ankan Ganguly and Kavita Ramanan. "Hydrodynamic limits of non-Markovian interacting particle systems on sparse graphs." Electronic Journal of Probability 29 (2024): 1-63.
>
> [3] Fabian et al. "Learning mean field control on sparse graphs." ICML 2025.

---

> > ### Author Rebuttal · Reviewer_XV6n · 2026-04-01
> >
> > My concerns have been addressed. I appreciate the additional work on the multi-agent setting, and I think even just a brief discussion of initial results or possibilities in the appendix would significantly strengthen the paper. I will raise my score to 5.

---

> > > ### Author Response · Authors · 2026-04-04
> > >
> > > We thank the reviewer for increasing the score. We agree that addressing the algorithmic aspect for the multi-agent setting at some detail would strengthen the paper, and we plan on doing the same.

---

### Official Review · Reviewer_xn8F · 2026-03-10

**Soundness:** 3
**Presentation:** 3
**Significance:** 4
**Originality:** 3
**Overall Recommendation:** 5
**Confidence:** 4

**Summary:**

This work provides exponentially close to optimal guarantees for m-hop policies in multi-agent MDPs with networked agents. This guarantee results from the m-hop local policies for each agent allowing the agents to act preemptively before the 1-step influences of agents beyond m-hops can influence the local state of the agent. These m-hop policies attain scalability improvements as they only depend on the m-hop neighbors rather than the states of all the agents. The multi-agent networked agents setting shows up in applications such as epidemic control, autonomous driving, wireless scheduling etc.

**Compliance With Llm Reviewing Policy:**

Affirmed.

**Final Justification:**

Overall, I recommend the paper for the reasons listed above. The authors have addressed my questions

**Key Questions For Authors:**

1) How limiting is the $\gamma C < 1$ assumption? I see a similar assumption was made by Zhang 2023. Is this weak influence assumption commonly held for applications?

2) I feel that a lower bound could be proved for this setting (that there exists a poor example that m hop policies are exponentially far away from optimal). Have you considered this and how difficult would it be to prove?

**Limitations:**

Yes.

**Strengths And Weaknesses:**

Soundness: 3 (Good)
Overall, this work appears to be technically sound. I did not catch any significant errors during my reading of the body of the paper or the proof of Theorem 1. Everything I read seems to be well supported.

Presentation: 3 (Good)
The submission is clearly written and easy to understand. The proofs in the appendix are easy to read. However, there are some mathematical typos that should be resolved.

Below are some that I spotted:

i) Theorem 4 looks identical to Theorem 2 (does not use H_i,j)

ii) The first set of equations in appendix A.2. appear to be missing $\gamma$ when taking the difference of bellman updates

iii) The notation $N_i^k$ and $N_k^i$ get flipped occasionally (see section A.3. Proof of Lemma 4 in the appendix)

Significance: 4 (Excellent)
The result of exponential decay of the m-hop policy has been sought after result, and has been a technical challenge in the networked RL space. It gives the impression of a simple and natural result but is quite difficult to prove. The work by Zhang 2023 [1] only achieves a polynomial rate.

Originality: 3 (Good)
The methods are novel and this is a very clear improvement over the previous approaches in that it achieves an exponential bound

[1] Zhang, Y., Qu, G., Xu, P., Lin, Y., Chen, Z., and Wierman, A. Global convergence of localized policy iteration in net-
worked multi-agent reinforcement learning. Proceedings of the ACM on Measurement and Analysis of Computing Systems, 7(1):1–51, 2023

---

> ### Author Rebuttal · Authors · 2026-03-31
>
> Thank you for your careful reading and comments.  Re the typos: thanks for catching, and we will correct in the revision.
>
> __Bounded spectral radius assumption:__ In section 5.3, we discuss an upper bound on the spectral radius $\rho(C) \leq d_{max} C_{max}$. This bound shows a natural trade-off between graph degree and interaction strength: higher connectivity requires weaker pairwise influences, and vice versa, for the condition to hold.
> In sparse graphs, for this condition to not hold, the transitions of each local state would have to become almost deterministic. Here is such an example. Consider the case where agents 1 to $n$ are arranged on a line, such that only agent 1 has a non-zero reward function, but all agents have actions.  It is possible to construct cases where, if all the transitions were to be deterministic, agent 1 needs to coordinate its action with agent $n$ to be optimal, and if it didn't, agent 1's local state goes into a permanent bad state such that the system incurs a very suboptimal reward.
>
> __Influence matrix condition in Zhang (2023):__ Zhang(2023) has a related assumption on $C$: they require $\|\|C\|\|_1 < 0.5$. This is a stronger condition than a bound on the spectral radius. The way this condition is used in Zhang (2023) is the following. They only consider policies where the action for each agent is purely a function of the states of their $m$-hop neighborhood. Note that this differs from our notion of $m$-hop cooperative policies as there is no coordination between the actions of nearby agents. For such policies, they use the bounded spectral radius condition to show that their associated $Q$-functions can be decomposed as a sum of $2m$-hop separable functions. The intuition here is that, since the action of any agent is a function of the states of their $m$-hop neighborhood, any change outside their $2m$-hop neighborhood will take at least $m$ hops to affect its local reward $R_i(S_i,a_i)$ at every point, and the propagation of these influences are controlled using the influence matrix $C$.
>
> They then use an entropy-regularized Bellman operator (with high regularization, see Theorem 1) to show that policy iteration is closed under these $m$-hop local policies. They require entropy regularization so that policy iteration generates a unique policy under the argmax. The high regularisation helps them decouple the policies as being $m$-hop local.
>
> We use the bounded spectral radius condition differently in our analysis. We use it to first show that the Bellman operator (without any regularization) is approximately closed under $m$-hop separable $Q$-functions. Secondly, we show that under repeated applications of the Bellman operator, the propagated errors can be controlled, thereby limiting the optimal $Q$-function to lie in a certain error ball around the class of $m$-hop separable $Q$-functions.
>
> __Lower bounds:__ You also raise an interesting question of proving \emph{lower bounds} on the performance of $m$-hop policies when this condition does not hold.  We suspect that this should be possible, because violation of it allows for large-range propagation of dependencies.  We will think about a simple lower bound construction along these lines --- thanks for the suggestion.

---

> > ### Author Rebuttal · Reviewer_xn8F · 2026-04-02
> >
> > Thank you for the detailed response! I will keep my original score

---

> > > ### Author Response · Authors · 2026-04-04
> > >
> > > We thank the reviewer for addressing our rebuttal. Do let us know if there are any other aspects of the submission that you would like addressed.

---

### Official Review · Reviewer_W1NX · 2026-03-12

**Soundness:** 3
**Presentation:** 3
**Significance:** 3
**Originality:** 3
**Overall Recommendation:** 5
**Confidence:** 3

**Summary:**

The authors study the problem of planning in a multi-agent setting with m-hop policies -- set of policies that a depend only on a local neighborhood of m steps. The focus on a social welfare setting, where they aim to maximize the cumulative reward across agents, and make a factorisation assumption on the transition model.

In general, even this structure may be result in NP-hard computational problems. The authors focus on the studying the performance gap between the optimal policy and an optimal m-hop policy, calculated by performing Bellman update. Specifically, this work provides upper bounds that bound the difference between these, depending on structural quantities of the MDP.

**Compliance With Llm Reviewing Policy:**

Affirmed.

**Key Questions For Authors:**

- See weaknesses of the paper.
- Can the authors provide any lower bound to the upper bounds introduces in this work?

**Limitations:**

Yes

**Strengths And Weaknesses:**

#Strengths

- Providing approximate algorithmic techniques for solving multi-agent systems is an interesting and open problem. The authors make progress towards that goal.
- The idea, to the best of my knowledge, is novel and may provide inspiration to other researchers.
- The paper is well written.

#Weaknesses

- The paper focuses only on the planning setting. It is not clear how to generalize these results to the learning setting.
- Experiments are quite basic.
- As the authors also suggest, solving the approximately optimal m-hop policy, is on its own, possibly, a hard problem.

---

> ### Author Rebuttal · Authors · 2026-03-31
>
> We thank the reviewer for their questions and comments.
>
> __Extension to the learning setting:__ To clarify, our results establish structural properties for the optimal Q-function and the optimal policy in graph-structured MDPs. In particular, we show that the optimal Q-function can be well-approximated by functions depending only on m-hop neighborhoods. This structural result naturally suggests an approximate Q-learning that is restricted to these local structures.
>
> Once a function class is fixed either for the Q-functions or for the policies, the sample complexity of RL is well-studied [1].  It usually scales with appropriate complexity measures of the function class (e.g., dimension, covering numbers). In finite state-action spaces, since the size of the state-action space dictates the complexity of the Q-function class, the sample complexity blows up exponentially in the number of agents in multi-agent settings. Thus, a consequence of our structural results is to show that using standard methods such as Q-learning, the sample complexity will only scale linearly in the number of agents.
>
> __Lower bounds:__ You also raise an interesting question of proving lower bounds on the performance of $m$-hop policies when this condition does not hold.  We suspect that this should be possible, because violation of it allows for large-range propagation of dependencies.  We will think about a simple lower bound construction along these lines --- thanks for the suggestion.
>
> [1] Azar et al. (2017). Minimax regret bounds for reinforcement learning.

---

> > ### Author Rebuttal · Reviewer_W1NX · 2026-04-03
> >
> > Dear authors,
> >
> > Thank you for your response. I will keep my current score and support the acceptance of this work.

---

> > > ### Author Response · Authors · 2026-04-04
> > >
> > > We thank the reviewer for addressing our rebuttal. Do let us know if there are any other aspects of the submission that you would like addressed.

---

### Official Review · Reviewer_rYam · 2026-03-21

**Soundness:** 2
**Presentation:** 1
**Significance:** 2
**Originality:** 2
**Overall Recommendation:** 3
**Confidence:** 3

**Summary:**

The paper studies cooperative MARL within graph-structured partially observable MDPs. It establishes conditions under which a globally optimal policy can be approximated by $m$-hop local policies, showing that the sub-optimality gap decays exponentially with respect to $m$.

**Compliance With Llm Reviewing Policy:**

Affirmed.

**Key Questions For Authors:**

- One minor concern is whether the setting is indeed meaningful for realistic scenarios, because the underlying graph is fixed. This means that the agents should always communicate with the same teammates throughout the whole learning/computation process. What happens if the graph changes either adversarially or via a known/unknown distribution?

**Limitations:**

N/A.

**Strengths And Weaknesses:**

Strengths:
- The paper studies a novel multi-agent MDP setting where the agents are embedded into an underlying graph. The setting uses the notion of the m-neighborhood to deal with the generally intractable POMDPs.
- For the single-agent setting, the paper not only proves the existence of an $m$-hop policy but also gives an efficient algorithm based on projected Q-iteration updates.

Weaknesses:
- While the authors provide an algorithm for the single-agent case, they have left the computation of the $m$-hop cooperative policy in the multi-agent setting as an open question. I believe that the paper would have been benefited a lot, if the authors had solved the multi-agent setting as well.
- One concern that I have is the assumption about the bounded spectral radius. The authors have tried to explain why this assumption is reasonable, however it remains unclear whether this assumption holds even for small MDPs. It would be beneficial that the authors provide toy examples showcasing whether this assumption holds and, in general, under what conditions.
- Last but not least, the writing should be improved. There are many typos and expressive language errors, including: 1. "Associated with each node is an agent, along with a local
state, a local action, and a local reward function that only
depends on its own state and action", 2. "Moreover, the state only on the
local neighborhood of state-action pairs.", 3. "epidemic con-
troml", 4. "co-ordination", 5. "PSACE-hard", 6. "m-neighorhood" (in the theorem statement), 6. in equation 6, \tilde{Q} has not been defined.

---

> ### Author Rebuttal · Authors · 2026-03-31
>
> We thank the reviewer for their careful reading and comments. We will first address the points raised in the Weaknesses section, and then answer the questions.
>
> Re typos:  Thank you for pointing out them out. We will make sure to proofread the paper more carefully in future versions.
>
> __Bounded spectral radius assumption:__ In section 5.3, we discuss an upper bound on the spectral radius $\rho(C) \leq d_{max} C_{max}$. This highlights that the key factor is not the size of the MDP, but rather the connectivity and interaction strength of the underlying graph. In particular, the condition can hold even for large systems, provided the graph is sufficiently sparse or the interactions are not too strong. This bound also reveals a natural trade-off between graph degree and interaction strength: higher connectivity requires weaker pairwise influences for the condition to hold.
> In sparse graphs, for this condition to not hold, the transitions of each local state would have to become almost deterministic. Here is a simple but illuminating example. Consider the case where agents 1 to $n$ are arranged on a line, such that only agent 1 has a non-zero reward function, but all agents have actions. It is possible to construct cases where, if all the transitions were to be deterministic, agent 1 needs to coordinate its action with agent $n$ to be optimal, and if it didn't, agent 1's local state goes into a permanent bad state such that the system incurs a very suboptimal reward.
>
> We can expand section 5.3 to add more such concrete examples, thanks for the suggestion.
>
> __Multi-agent setting:__ We agree that algorithmic guarantees for this setting is an important facet of this problem. Our current result (Theorem 2) establishes the existence and the structure of near-optimal m-hop cooperative policies, which already goes beyond the scope of prior work.
>
> Since the submission, we have made progress on the algorithmic side. We consider a projected Bellman iteration using the projection operator defined in Appendix B.2. We show that this projected Bellman iteration converges exponentially fast to a ball around $Q^*$, with the radius of the ball decaying exponentially in $m$. This result requires a stronger version of the spectral radius bound, i.e., $\sqrt{\gamma} n \rho(C)<1$, where $n$ is the number of players. The dependence on $n$ arises due to the need for controlling the projection errors of each of the $n$ players. It's not clear to us if we have the space to add this result to the current submission, but if the reviewers think it necessary, we would be happy to oblige or provide more details.
>
> __Dynamic graphs:__ In the proofs of our results, the main quantity that would change if the underlying graph is dynamic, is the influence matrix $C$, which would have a time dependence $\{C_t\}$. At a technical level, we only require control on matrix sums of the form $\sum_{k \geq 0} \gamma^k \prod_{t \leq k} C_t $, which is possible for instance with uniform control on $\rho(C_t)$. Nothing would change on the algorithmic side!

---

> > ### Author Rebuttal · Reviewer_rYam · 2026-04-02
> >
> > I would like to thank the authors for their response. I believe that the new results on the multi-agent setting are important, especially since the authors claim in the abstract to "study a cooperative form of multi-agent reinforcement learning". While including these new results certainly makes the submission much stronger, it is important that such substantial additions undergo a full peer review. Therefore, I will maintain my score.

---

> > > ### Author Response · Authors · 2026-04-04
> > >
> > > We thank the reviewer for addressing our rebuttal. Do let us know if there are any other aspects of the submission that you would like addressed.

---

### Decision · Program_Chairs · 2026-04-30

**Decision:**

Accept (regular)

**Comment:**

Summary: The paper investigates graph-structured Markov Decision Processes to determine when global optimal policies can be effectively approximated by $m$-hop local policies. The authors provide a theoretical foundation for this approximation by controlling influence propagation through a Dobrushin-type stability matrix, showing that the approximation error decays exponentially with distance.

Decision: Despite the weakness of the limited depth in the multi-agent algorithmic discussion and persistent typos throughout the manuscript, the paper has strong theoretical and conceptual contributions. Please ensure these multi-agent details (see discussion with rYam) are expanded and all grammatical errors are corrected in the camera-ready version. Among the reviewers, there is significant enthusiasm for the paper to be accepted and the AC agrees.